Technical Report

# Family-based genome-wide association study designs for increased power and robustness

Junming Guan ®[1] ✉, Tammy Tan[2], Seyed Moeen Nehzati[1,3], Michael Bennett ®[2], Patrick Turley ®[4,5], Daniel J. Benjamin ®[1,2,6] & Alexander Strudwick Young ®[1,6] ✉

Family-based genome-wide association studies (FGWASs) use random, within-family genetic variation to remove confounding from estimates of direct genetic effects (DGEs). Here we introduce a 'unified estimator' that includes individuals without genotyped relatives, unifying standard and FGWAS while increasing power for DGE estimation. We also introduce a 'robust estimator' that is not biased in structured and/or admixed populations. In an analysis of 19 phenotypes in the UK Biobank, the unified estimator in the White British subsample and the robust estimator (applied without ancestry restrictions) increased the effective sample size for DGEs by 46.9% to 106.5% and 10.3% to 21.0%, respectively, compared to using genetic differences between siblings. Polygenic predictors derived from the unified estimator demonstrated superior out-of-sample prediction ability compared to other family-based methods. We implemented the methods in the software package snipar in an efficient linear mixed model that accounts for sample relatedness and sibling shared environment.

Genome-wide association studies (GWASs) have identified thousands of associations between genetic variants and human phenotypes[1]. Standard GWAS estimates the association between a phenotype and an allele by regression of individuals' phenotypes onto the number of copies of the allele that they carry, with some adjustment for covariates. Multiple phenomena contribute to the associations, which we call 'population effects', as they reflect the genotype-phenotype association in the population[2–5]: causal effects of alleles (both of the tested variant and those in linkage disequilibrium (LD) with the tested variant) carried by the individual on the individual, called direct genetic effects (DGEs); effects of alleles in relatives through the environment, called indirect genetic effects (IGEs) or genetic nurture[6]; and effects of other genetic and environmental factors that the tested variant is correlated with due to population stratification and assortative mating (AM)[3,4,6–10]. Biased sampling can also affect population effect estimates[11].

If we consider the goal of GWAS to be estimation of DGEs, then the other contributing factors can be considered as confounds. Adjustment for genetic principal components (PCs) and linear mixed models (LMMs) reduces confounding due to population stratification[7,8] and AM[12], but residual confounding often remains[3,8,9,12]. The consequences of this include (1) biased estimates of heritability and the traits' shared genetic architectures (through genetic correlation estimates)[13–16]; (2) biased inferences from Mendelian randomization[17]; (3) bias in polygenic indices (PGIs, also called polygenic scores) that may contribute to the drop in predictive accuracy when predicting across genetic ancestries[18]; and (4) biased inferences of natural selection[9,16,19].

Family-based GWAS (FGWAS) adds parental genotypes to the regression used in GWAS (Methods). FGWAS thereby uses variation due to random segregations of genetic material during meiosis to estimate DGEs, removing confounding due to gene-environment correlation

[1]UCLA Anderson School of Management, Los Angeles, CA, USA. [2]National Bureau of Economic Research, Cambridge, MA, USA. [3]Department of Economics, New York University, New York, NY, USA. [4]Department of Economics, University of Southern California, Los Angeles, CA, USA. [5]Center for Economic and Social Research, University of Southern California, Los Angeles, CA, USA. [6]Department of Human Genetics, UCLA David Geffen School of Medicine, Los Angeles, CA, USA. ✉e-mail: junm.guan@gmail.com; alextisyoung@gmail.com

and nonrandom mating[3,16]. However, requiring both parents' genotypes limits the sample to which FGWAS can be applied. An alternate approach, which we call 'sib-differences', uses genetic differences between siblings to estimate DGEs[16] (Methods), enabling use of samples with genotyped siblings but without genotyped parents to be used. However, sib-differences has lower power than FGWAS when parental genotypes are available[3].

Young et al.[3] proposed an alternative approach that could be applied to sibling data: treat parental genotypes as missing data and impute them according to Mendelian laws, and then use the imputed parental genotypes in place of the observed ones (Methods). Provided that the imputation is unbiased, the DGE estimates are unbiased and consistent[3]. This approach increases the effective sample size for DGEs by up to one-third compared to sib-differences. (The relative effective sample size of estimator $a$ compared to estimator $b$ is the ratio of the sampling variance of estimator $b$ to estimator $a$; a relative effective sample size above 1 indicates greater power for estimator $a$ compared to estimator $b$.) The imputation method enables the inclusion of any genotyped sample with at least one genotyped first-degree relative, including samples with one or both parent(s) genotyped but without genotyped siblings, further increasing power[3].

However, the Young et al. method ignores most of the sample in datasets like the UK Biobank (UKB), where only ~10% have a genotyped first-degree relative[20]. Samples of individuals without genotyped first-degree relatives (hereafter 'singletons') can provide precise estimates of $\beta$, the population effect, as from standard GWAS. Under random mating[3], $\beta = \delta + \alpha$, where $\delta$ is the DGE and $\alpha$ is the average coefficient on the parents' genotypes (Methods)—called the average nontransmitted coefficient (NTC). A precise estimate of the population effect therefore puts a constraint on the set of plausible values the FGWAS parameters, $\delta$ and $\alpha$, can take.

Following this intuition, we develop an FGWAS estimator that has increased power by including singletons through imputation. However, we show that strong population structure leads to bias in the DGE estimates. We also develop an estimator that is robust to population structure and admixture for use in genetically diverse samples. This estimator is more powerful than sib-differences because it includes samples with one or both parents genotyped but without genotyped siblings, and it uses parental genotypes when available for samples with genotyped siblings. We examine the estimators in simulations with different levels of population structure, enabling researchers to choose the appropriate analysis depending on their data. We demonstrate increased power for estimation of DGEs in the UKB and in out-of-sample PGI prediction in the Millennium Cohort Study (MCS).

## Results
### Including singletons in FGWAS
We extend the imputation method described in Young et al.[3] to singletons. We observe two out of four parental alleles in a singleton's genotype—as in a sibling pair that have inherited the same alleles from both mother and father, which is expected for one-quarter of the genome[3]. The two missing parental alleles are imputed using the allele frequency, resulting in imputed parental genotypes that are linear functions of the singletons' genotypes (Methods).

Consider that we have a genotyped and phenotyped sample partitioned into two disjoint subsets: a subset with at least one genotyped first-degree relative (which we call the 'related sample'), where missing parental genotypes have been imputed as in Young et al.[3]; and a singleton sample, where parental genotypes have been imputed linearly. The estimator that we propose, called the 'unified estimator', uses the imputed parental genotypes when they are not observed, including for the singleton sample (Fig. 1a,b and Methods).

In Supplementary Note 2.1, we derive theoretical results on the gain in effective sample size for DGEs from including singletons. Consider the case where we have $n_0$ independent sibling pairs whose parental genotypes are imputed using phased data as in Young et al.[3], and we add $n_1$ singletons with their parental genotypes linearly imputed. Assuming that siblings' phenotypes are uncorrelated conditional on the regression covariates (that is the regression residuals are uncorrelated), adding $n_1$ singletons gives an effective sample size $1 + \frac{n_1}{2(3n_0+n_1)}$ times higher than using only the $n_0$ sibling pairs (Supplementary Note 2.1.1). The theoretical gain in effective sample size converges to 50% as $n_1/n_0 \to \infty$. Imputation as in Young et al.[3] already gives a gain of up to one-third; thus the effective sample size of the unified estimator can be up to twice as large as the sib-difference estimator, and can be even higher when samples with genotyped parents are also available. The gain in effective sample size declines with the correlation between the siblings' residuals and, when imputing from siblings without phased data, with minor allele frequency (Fig. 1c,d).

We derive equivalent results for adding $n_1$ singletons to $n_0$ samples with one parent genotyped, where the missing parent's genotype has been imputed using phased data as in Young et al.[3]. The effective sample size for DGEs is approximately $1 + \frac{n_1}{(3n_1+4n_0)}$ times higher than using the parent-offspring pairs alone, converging to 4/3 as $n_1/n_0 \to \infty$.

One can obtain an estimate of the standard GWAS population effect, $\beta$, by $\hat{\beta} = \hat{\delta} + \hat{\alpha}$, where $\hat{\delta}$ and $\hat{\alpha}$ are the DGE and average NTC estimates from the unified estimator. By performing the analysis using all genotyped samples that would normally be used in a standard GWAS (Fig. 1a,b), one obtains an estimate of $\beta$ almost identical to that from standard GWAS (correlation 0.998; Extended Data Fig. 1). Thus, by including singletons via linear imputation, we unify FGWAS and standard GWAS in one analysis.

We estimated DGEs with the unified estimator for the simulated phenotypes from Young et al.[3], which simulated scenarios including AM and IGEs (Supplementary Note 1). We found the unified estimator increased the effective sample size for DGEs compared to using only the related sample and did not introduce any detectable bias.

### Population-structure-robust estimator
The imputation proposed by Young et al.[3] uses the allele frequency to impute unobserved parental alleles, becoming biased when there is population structure as it does not account for variation in allele frequencies across subpopulations[3]. Young et al. showed that, in an island model of population structure, the estimator of DGEs from sibling pairs with parental genotypes imputed from phased data tends to $\hat{\delta} = \delta + c\alpha$, where $c$ is a function of Wright's $F_{st}$ (the proportion of variation at a locus due to between-population differences in allele frequencies). When $F_{st}$ is small, $c \approx F_{st}/2$, implying the bias, $c\alpha$, will be negligible for European genetic ancestry samples, where $F_{st}$ has been estimated to be on the order of $10^{-3}$ (ref. 21). In contrast, standard GWAS estimates $\beta = \delta + \frac{1+3F_{st}}{1+F_{st}}\alpha$.

Here we develop two estimators that maximize power for estimation of DGEs while being robust to population structure (Methods). We generalize an estimator proposed in Young et al. by partitioning the sample based on which parental alleles that were not transmitted to the focal, phenotyped individual (proband) we have observed: this gives four groups (Table 1 and Extended Data Fig. 2) depending on whether we have observed one or both nontransmitted (NT) parental alleles, and if only one has been observed, whether the NT allele is from the mother, father or unknown. We call this estimator the 'nontransmitted (NT) estimator'. We prove that this estimator gives consistent estimates of DGEs under an island model of population structure (Supplementary Note 2.3.2). However, the NT estimator can give biased estimates when there are differences in allele frequencies between mothers and fathers (Supplementary Note 2.3.3), as in recently admixed samples.

We therefore developed the 'robust estimator' that uses only the random variation in offspring genotype given parental genotype, which is the principle underlying the properties of FGWAS with fully observed parental genotypes. This estimator, like the NT estimator, partitions the sample based on which NT alleles have been observed

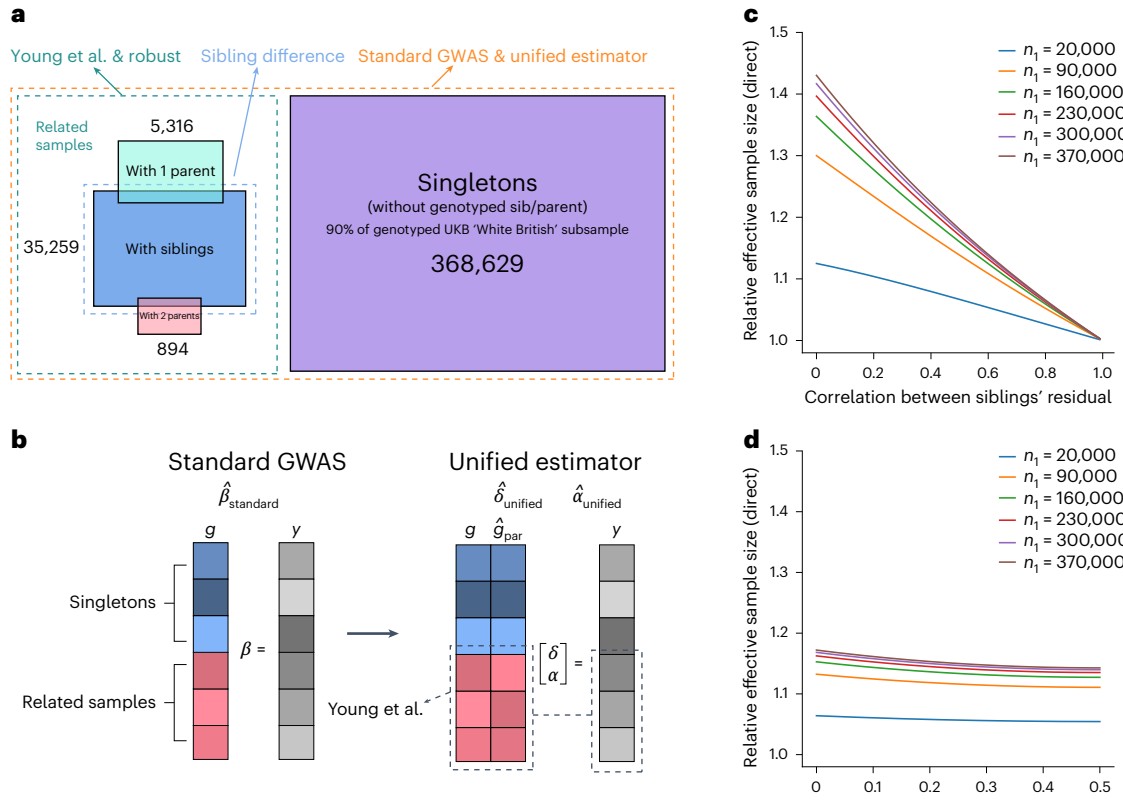

**Fig. 1 | Illustration of different standard and FGWAS estimators and theoretical gain in effective sample size for DGEs. a**, We illustrate the different sample subsets used by different FGWAS and standard GWAS methods. We give the numbers for each subset for the UKB 'White British' sample for illustration. The sibling difference estimator uses samples with one or more siblings' genotypes observed (35,259 individuals), whereas the Young et al. estimator uses all related samples, which also include individuals with both parents' genotypes observed (894) and those with one parent's genotype observed (5,316); in addition to the related samples, the standard GWAS and unified estimators also use singletons (368,629). **b**, Illustration of regressions performed by standard GWAS and the unified estimator. Through linear imputation of parental genotypes, the unified estimator incorporates singletons into the FGWAS regression, enabling use of the same sample as standard GWAS to estimate the

parameter vector $[\delta, \alpha]^T$. Although the design matrix for the singleton subset (in blue) in FGWAS is collinear, the design matrix for the related sample subset (in red) is not, so the stacked design matrix is not collinear. **c**,**d**, We show the effective sample size for the unified estimator applied to $n_0 = 20,000$ sibling pairs and $n_1$ singletons, relative to the effective sample size of using the sibling pairs alone with imputation. The parental genotypes in the sibling sample are imputed[3] using phased data (**c**) and unphased data (**d**). The parental genotypes for the singletons are imputed linearly. The theoretical gain depends upon the correlation between the siblings' residuals, which we show in **c**. When imputing using unphased data, the gain depends upon the minor allele frequency[3], which we show in **d** for a fixed correlation between siblings' residuals of 0.3. We confirmed the theoretical results using simulations (Supplementary Note 2.1).

(Table 1). However, it differs in the regressions it performs. The robust estimator performs uniparental regressions for samples with one NT allele of known parent-of-origin observed; for example, for a sample with a mother genotyped, the regression is performed on the maternally transmitted allele and the mother's genotype, thereby only using the random variation in maternally inherited allele given maternal genotype to estimate the DGE.

The advantage of the robust estimator over using only sib-differences and/or samples with both parents genotyped is that it enables optimal use of samples with a single parent genotyped while not using any allele frequency information that can introduce bias in structured populations. When only siblings are genotyped with no parents, it becomes equivalent to sib-differences (Supplementary Note 2.5.1); and when all samples have both parents genotyped, it becomes FGWAS with fully observed parental genotypes.

### Comparison of estimators in simulated populations

We examined the power (measured by effective sample size) and bias of the different estimators (Table 2) in simulations with different levels of population structure, as measured by Wright's $F_{st}$. We simulated populations of 2,000 independent sibling pairs and 18,000 independent

singletons, mimicking UKB data proportions. We considered two simulation setups: (1) two equally sized subpopulations with ancestral allele frequencies equal to 0.5; and (2) 100 subpopulations with ancestral minor allele frequencies, $f$, drawn from a distribution with density proportional to $1/f$ for $0.05 < f < 0.5$. In both scenarios, the allele frequencies in the subpopulations were drawn from the Balding-Nichols distribution for $F_{st}$ set at 0, 0.001, 0.01, or 0.1: $F_{st} = 0.001$ is roughly the level of differentiation between neighboring European populations[21], and $F_{st} = 0.1$ is roughly the level of differentiation between European and East Asian ancestry populations[22]. The phenotypes were simulated without any causal genetic effects but with subpopulation membership explaining 50% of the variance (Methods).

The bias from population stratification confounding is due to the correlation between the subpopulation allele frequencies and the subpopulation phenotype means (Methods). Because allele frequencies and phenotype means were sampled independently, the bias for an individual single-nucleotide polymorphism (SNP) has expectation zero (across repeated simulations) but has non-zero variance across SNPs (and repeated simulations). The magnitude of population stratification confounding can therefore be evaluated by the nonsampling variance—the variance in the estimates not explained by sampling

**Table 1 | Groups and regressions for the NT and robust estimators**

| Group | Example genotype data types | NT alleles observed | NT estimator regression | Robust estimator regression |
|---|---|---|---|---|
| Maternal NT | Mother-child pairs<br>Mother and sibling pair in IBD2 | Maternal | $y_{ij} \sim g_{ij} + \hat{g}_{p(i)} + g_{m(i)}$ | $y_{ij} \sim g_{ij}^m + g_{m(i)}$ |
| Paternal NT | Father-child pairs<br>Father and sibling pair in IBD2 | Paternal | $y_{ij} \sim g_{ij} + g_{p(i)} + \hat{g}_{m(i)}$ | $y_{ij} \sim g_{ij}^p + g_{p(i)}$ |
| Both NT | Sibling pairs in IBD0<br>Parent-offspring trios | Paternal and maternal | $y_{ij} \sim g_{ij} + g_{par(i)}$ | $y_{ij} \sim g_{ij} + g_{par(i)}$ |
| One NT | Sibling pairs in IBD1 without genotyped parents | Paternal or maternal | $y_{ij} \sim g_{ij} + \hat{g}_{par(i)}$ | $y_{ij} \sim g_{ij} + \bar{g}_{sib(i)}$ |

We partition the sample with at least one NT parental allele observed into four groups (Extended Data Fig. 2), perform separate regressions in each group and meta-analyze the resulting DGE estimates (Methods). We show that the NT estimator is robust to an island model of population structure, but not to admixture, whereas the robust estimator is robust to both (Supplementary Notes 2.3 and 2.4). For the regression column, $y_{ij}$ is the phenotype of sibling $j$ in family $i$; $g_{ij}$ the corresponding genotype; $g_{ij}^m$ and $g_{ij}^p$ are the maternally and paternally transmitted alleles; $g_{p(i)}$ and $g_{m(i)}$ are the paternal and maternal genotypes; $g_{par(i)} = g_{p(i)} + g_{m(i)}$; a caret indicates a genotype that has been imputed from phased data as in Young et al.[3]; for example, $\hat{g}_{par(i)}$ refers to the imputed sum of parental genotypes. $\bar{g}_{sib(i)}$ is the mean genotype among all siblings in family $i$. (IBD0 is when siblings share no alleles by descent from their parents, IBD1 is when siblings share one allele by descent from their parents and IBD2 is when siblings share both alleles by descent from their parents.)

**Table 2 | Summary of estimators**

| Estimator | Data types used | Procedure | Sample size in UKB |
|---|---|---|---|
| Sibling difference | Genotyped and phenotyped samples with at least one genotyped sibling | Regression of sibling phenotype differences onto sibling genotype differences, or regression onto deviation of sibling genotype from sibship mean[16] | 35,259 (White British)<br>46,698 (all ancestry) |
| Robust | Genotyped and phenotyped samples with at least one observed NT parental allele (Extended Data Fig. 2) | Perform separate regressions (Table 1) in each group and perform an inverse-variance-weighted, fixed-effects meta-analysis of DGE estimates | 44,570 (White British)<br>51,875 (all ancestry) |
| NT | Genotyped and phenotyped samples with at least one observed NT parental allele (Extended Data Fig. 2) | Perform separate regressions (Table 1) in each group and perform an inverse-variance-weighted, fixed-effects meta-analysis of DGE estimates | 44,570 (White British)<br>51,875 (all ancestry) |
| Young et al. | Genotyped and phenotyped samples, with genotyped first-degree relatives, in a homogeneous ancestry group | Fit FGWAS Model 1 or 2 (Methods) using imputed and/or observed parental genotypes | 44,570 (White British) |
| Unified | Genotyped and phenotyped samples, with or without relatives, in a homogeneous ancestry group | Fit FGWAS Model 1 or 2 (Methods) using imputed and/or observed parental genotypes | 408,254 (White British) |
| Standard GWAS | Genotyped and phenotyped samples, with or without relatives, in a homogeneous ancestry group | Regress proband genotypes on proband phenotypes | 408,254 (White British) |

The robust and NT estimators differ in the regressions they perform in each group (Table 1). See also Fig. 1 and Extended Data Fig. 2.

error, which must be due to population stratification bias, as there are no causal effects (Methods). We measure this relative to the nonsampling variance for standard GWAS and $F_{st} = 0.001$, comparable to the level of stratification bias in a standard GWAS in a homogeneous ancestry sample. (We also give the mean $Z^2$ statistic—which should be 1 under the null—a common measure of test-statistic inflation in GWAS. However, the nonsampling variance provides a fairer comparison of levels of bias, as mean $Z^2$ is also affected by sampling variance, which varies across estimators.)

For the two-subpopulation setup (Fig. 2), the sibling difference/robust—here the robust estimator reduces to the sib-difference estimator, as no parental genotypes are observed (Supplementary Note 2.5.1)—and the NT estimators have no detectable bias from population stratification for any level of $F_{st}$ (Fig. 2a–d), and the standard GWAS estimator has the most bias (Fig. 2a,c), with statistically significant bias for $F_{st} \geq 10^{-3}$. The unified and Young et al. estimators do not have detectable bias except for $F_{st} = 0.1$, with the unified estimator having greater bias than the Young et al. estimator (Fig. 2a–d). This result is expected because the unified estimator includes a large sample of singletons, for which the two unobserved parental alleles are imputed using the overall allele frequency, leading to bias in a structured population.

In the 100 subpopulation setup (Fig. 3), we also included standard GWAS with adjustment for 20, 50 and 99 inferred genetic PCs (Methods). (Because there are 100 subpopulations, 99 PCs should be sufficient to separate all subpopulations if inferred correctly[7,23].) Unlike in the two subpopulation setup, we did not find statistically significant evidence ($P < 0.05$) of bias for any of the family-based estimators.

This is likely to be because the magnitude of population stratification confounding goes down with the number of subpopulations (Methods).

However, we found statistically significant evidence of bias for standard GWAS when $F_{st} > 0$ regardless of how many PCs we controlled for, with one exception: when $F_{st} = 0.1$ and we controlled for 99 PCs. The likely reason is that it is difficult to infer 99 PCs correctly without very large sample sizes when population structure is subtle ($F_{st} \leq 0.01$) but becomes easier when population structure is stronger (as in the $F_{st} = 0.1$ scenario). This is related to the known phase transition whereby it becomes possible to accurately infer latent factors (for example, subpopulation membership) that structure random matrices (for example, SNP genotype matrices) once the sample size passes a certain threshold, depending on the strength of those latent factors[7]. In real-world genetic data, population structure exists on multiple scales, reflecting both recent and ancient structure, with genetic PCs only partly capturing subtle and recent structure[2,10].

### Bias–variance tradeoff for different estimators

We compare the estimators in a bias-variance framework (Fig. 4 and Extended Data Figs. 3 and 4) based on the two subpopulation simulations. The Young et al. estimator boosts power compared to the sib-difference, robust and NT estimators but introduces a slight bias due to population structure that becomes detectable for $F_{st} = 0.1$. The unified estimator adds singletons, gaining power at the cost of increased bias due to population structure, but this only becomes apparent when $F_{st} = 0.1$. The standard GWAS estimator has greater effective sample size

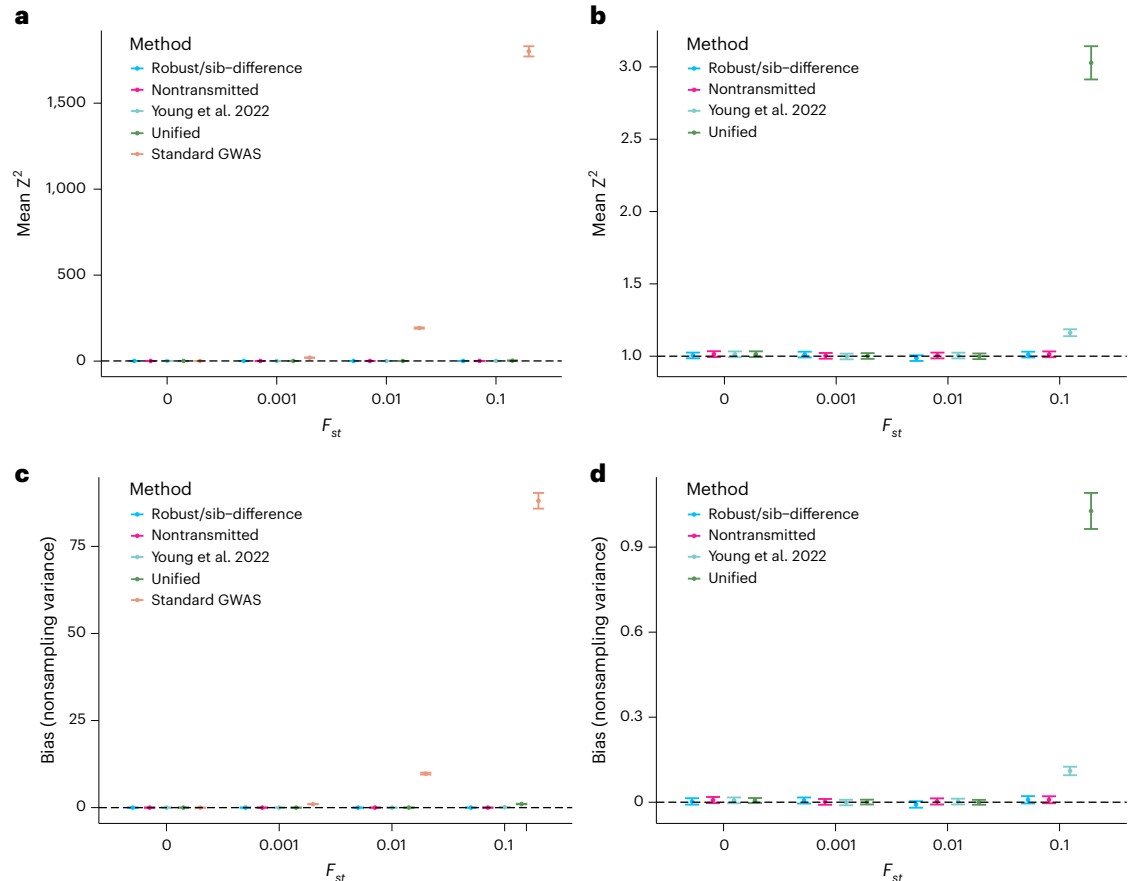

**Fig. 2 | Bias and nonsampling variance of GWAS estimators for different levels of population structure.** We simulated four different populations with different levels of structure, as measured by Wright's $F_{st}$. Each population consisted of two equally sized subpopulations with 2,000 independent sibling pairs and 18,000 singletons. Allele frequencies for the two subpopulations were simulated from the Balding-Nichols[31] model with ancestral allele frequency set to 0.5 (Methods). We simulated phenotypes with no causal genetic effects but where subpopulation membership explained 50% of the phenotypic variance, so that any deviation from the null distribution indicates population stratification confounding. **a**, Mean of squared Z-statistics across 20,000 SNPs for the four estimators, which are expected to be above 1 (dashed line) when there is bias due to population stratification. **b**, Same as **a** but with the standard GWAS removed. **c**, Mean of nonsampling variances (Methods) of the estimators relative to that observed for standard GWAS with $F_{st}$ = 0.001, which gives a measure of the magnitude of bias due to population stratification, with values above 0 indicating bias. **d**, Same as **c** but with the standard GWAS removed. Error bars display 95% jackknife confidence intervals over 20,000 SNPs.

than the other methods but at the cost of much greater bias in structured populations (Extended Data Fig. 3), in addition to other biases (such as from IGEs and AM[3]) that were not simulated here.

The simulation results show that the unified estimator has the greatest power out of the family-based estimators but shows bias when there is strong structure ($F_{st}$ > 0.01). The NT estimator is the most powerful estimator that is robust to structure but is vulnerable to confounding when there has been recent admixture (Supplementary Note 2.3.3). Given the small difference in power between the robust and NT estimators, we recommend the robust estimator for strongly structured samples unless recent admixture can be conclusively ruled out.

## LMM accounting for sample relatedness
We developed an LMM that includes random-effects specified by both sibship and a sparse genetic relatedness matrix (GRM), which is fast enough to perform genome-wide analyses in biobank-scale datasets while accounting for genetic relatedness and sibling shared environment (Methods). We give example runtimes in Supplementary Table 2.

## Application of estimators to UKB
We applied the estimators to 19 phenotypes using UKB data (Methods and Table 2). We applied the Young et al. and unified estimators to

the White British subsample (Fig. 1). We applied the sib-difference estimator to the sample with at least one genotyped sibling, and the robust estimator to the sample with at least one genotyped first-degree relative. No ancestry restrictions were applied for the robust and sib-difference estimators (Table 2)—although the resulting sample was 85.9% White British, it covered most of the genetic diversity captured by the first two PCs of 1000 Genomes[24] genotype data (Extended Data Fig. 6).

We compared the estimators' effective sample sizes (Fig. 5). The gain in effective sample size over sib-differences declined with the phenotypic correlation between siblings, as expected from theory[3] (Fig. 1c,d). Across the 19 phenotypes, the unified estimator had an effective sample size between 24.5% (height) and 42.6% (number of children in males) higher than the Young et al. estimator (Supplementary Table 1). As the Young et al. estimator already gains between 18.0% (height) and 45.3% (subjective well-being), this implies the unified estimator gains between 46.9% (height) and 106.5% (subjective well-being) over sib-differences. By not imposing ancestry restrictions, the robust estimator uses a larger sample (51,875) than the Young et al. estimator (44,570). The robust estimator also uses a larger sample than the sib-difference estimator (46,698) due to inclusion of samples without genotyped siblings, gaining between 10.3%

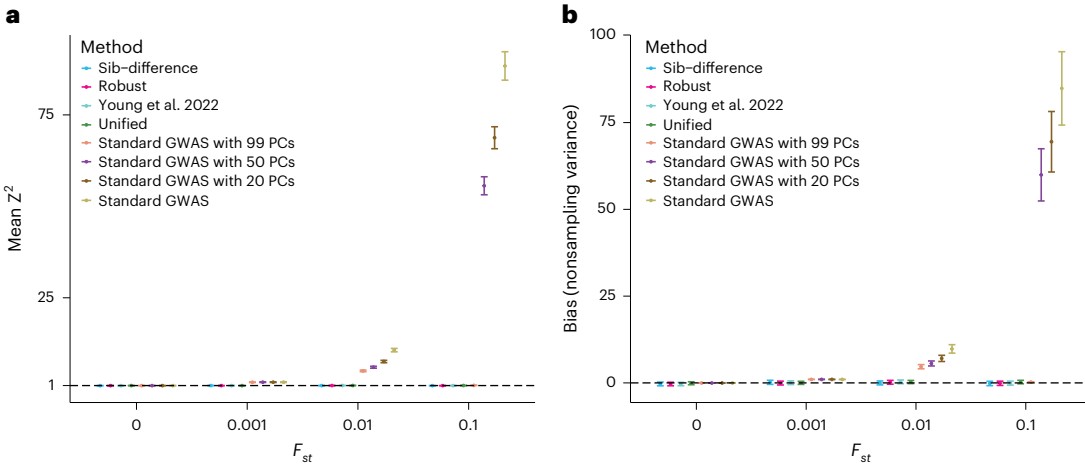

**Fig. 3 | Bias and nonsampling variance of estimators under complex population structure.** We simulated four different populations with different levels of structure, as measured by Wright's $F_{st}$. Each population consisted of 100 equally sized subpopulations with 100 independent sibling pairs and 900 singletons. Allele frequencies for the subpopulations were simulated from the Balding-Nichols[31] model with ancestral allele frequencies, $f$, drawn from a distribution proportional to $1/f$ (Methods). We simulated phenotypes with no causal genetic effects but where subpopulation membership explained 50% of the phenotypic variance, so that any deviation from the null distribution indicates population stratification confounding. For standard GWAS estimators, we inferred PCs and performed standard GWAS adjusting for different numbers of PCs (Methods): 0, 20, 50 and 99. **a**, Mean of squared Z-statistics across 4,000 SNPs for the four estimators, which is expected to be above 1 (dashed line) when there is bias due to population stratification. **b**, Mean of nonsampling variances (Methods) of the estimators relative to the that observed for standard GWAS with $F_{st} = 0.001$, which gives a measure of the magnitude of bias due to population stratification, with values above 0 indicating bias. Error bars display 95% jackknife confidence intervals over 4,000 SNPs.

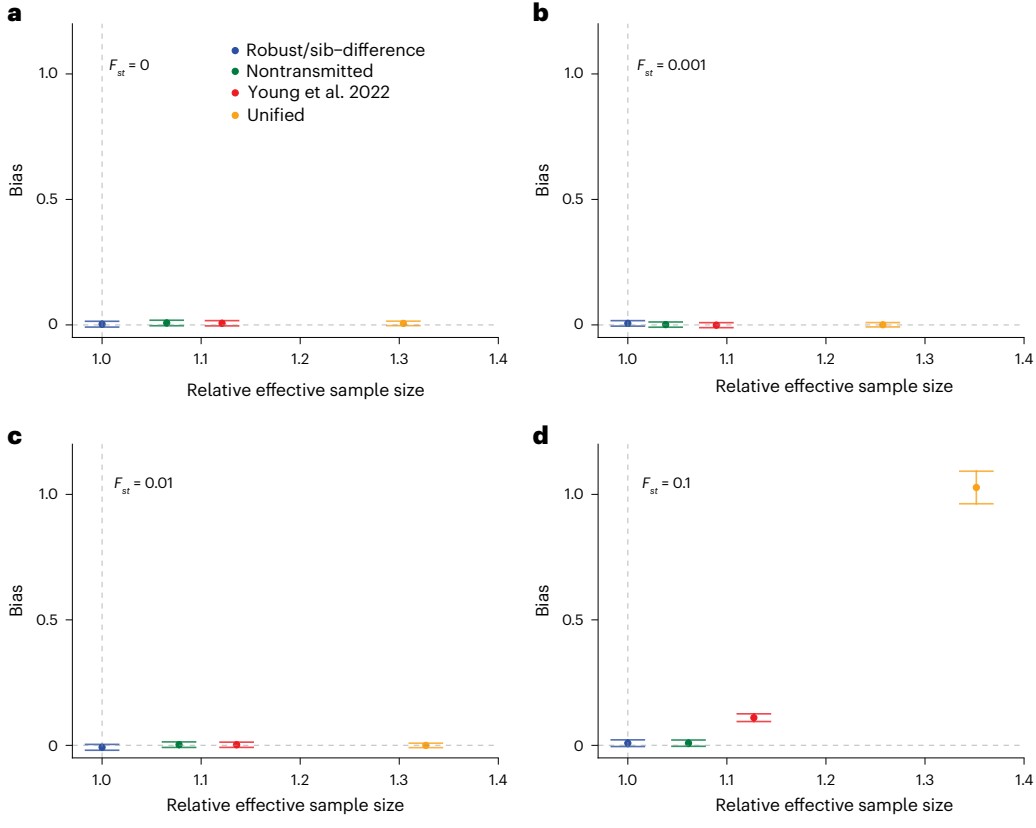

**Fig. 4 | Bias-variance tradeoff for family-based estimators. a–d**, The simulated datasets used in Fig. 2 are used for this demonstration: 2,000 independent sibling pairs and 18,000 singletons in each of two subpopulations with different levels of $F_{st}$ (Methods): $F_{st} = 0$ (**a**), $F_{st} = 0.001$ (**b**), $F_{st} = 0.01$ (**c**) and $F_{st} = 0.1$ (**d**). The effective sample size (*x*-axis) is defined relative to that of the sib-difference estimator (Table 2) and should be equal to 1 (vertical dashed line) for the robust/ sib-difference estimators—which are equivalent here—and higher than 1 for the other estimators. Bias (*y*-axis) is measured as the nonsampling variance (Methods) relative to that for standard GWAS with $F_{st} = 0.001$, and is expected to be above 0 (horizontal dashed line) when there is bias due to population stratification. Error bars display a 95% jackknife confidence interval over 20,000 SNPs. See Extended Data Figs. 3 and 4 for plots including the standard GWAS estimator and a sibling-only scenario (that is, no singletons).

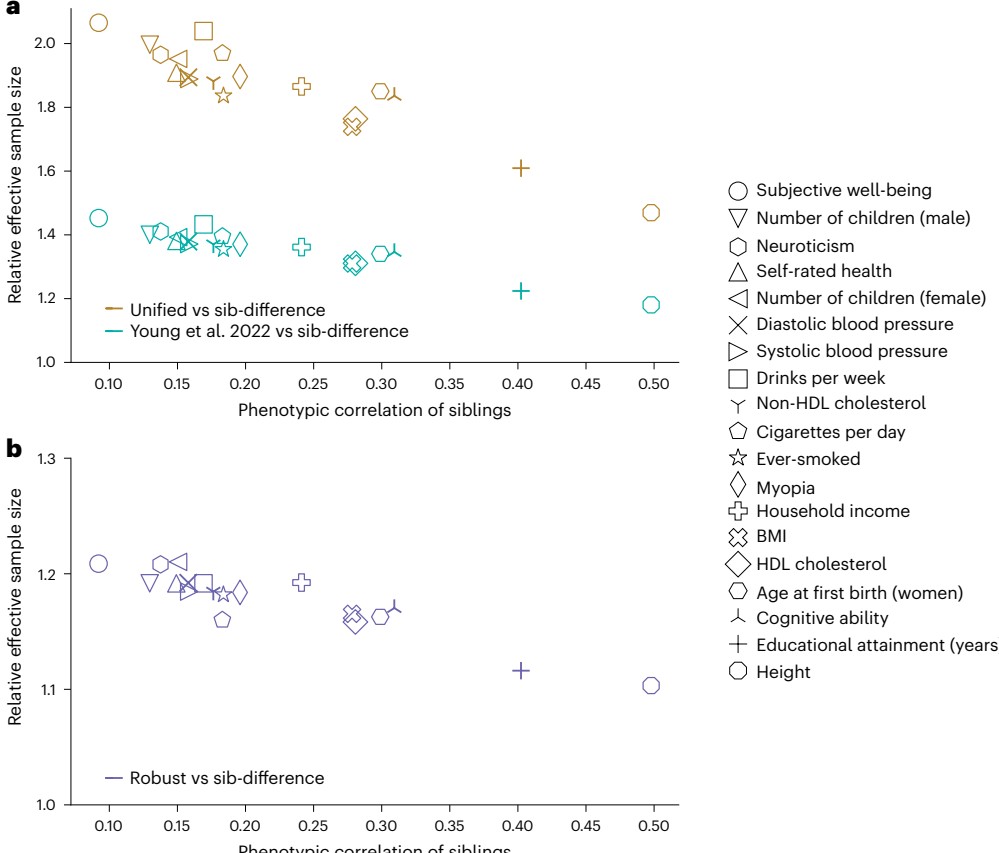

**Fig. 5 | Empirical gain in effective sample for DGEs.** We compute the effective sample size of the different estimators in UKB data (Table 2 and Supplementary Table 1) relative to that of the sib-difference estimator ($y$-axis), so that a value of ($1 + y$) means a gain of 100y% in effective sample size over the sib-difference estimator (Methods). We give the phenotypic correlation between siblings on the $x$-axis, as theory indicates the gain in effective sample size should decline with this correlation (Fig. 1c). **a**, Effective sample size for the unified (actual $n = 408,254$) and Young et al. (actual $n = 44,570$) estimators relative to the sib-difference estimator (actual $n = 35,259$) within the White British ancestry subsample. **b**, Effective sample size for the robust estimator (actual $n = 51,875$) relative to sib-difference estimator (actual $n = 46,698$), applied to the relevant samples without ancestry restrictions. The Young et al. estimator is more powerful than the sib-difference estimator because it uses information on NT parental alleles inferred by Mendelian imputation[3], and because it can incorporate individuals with one or both parents genotyped but without any siblings genotyped. The unified estimator gains over the Young et al. estimator by further including individuals without any genotyped first-degree relatives (singletons) through linear imputation (Fig. 1a,b). The robust estimator gains power over the sib-difference estimator by using parental genotypes for samples with one or both parents genotyped (Methods). HDL, high-density lipoprotein.

(height) and 21.0% (number of children, female) in effective sample size (Supplementary Table 1).

**Polygenic prediction in the MCS**

We evaluated the performance of the different estimators for out-of-sample prediction of height, body mass index (BMI) and general certificate of secondary education (GCSE) grades (a measure of educational achievement) using PGIs derived from DGE estimates (DGE PGIs) and population effect estimates (Methods). We found that all estimators yielded PGIs statistically significantly correlated with their respective phenotypes in the European ancestry (hereafter 'EUR') sample (Fig. 6a and Supplementary Table 4). Population-effect PGIs were substantially more predictive than DGE PGIs, as expected from the larger effective sample size of population effect estimates (Extended Data Fig. 3). Out of the DGE PGIs, the unified estimator gave the best predictions for BMI and GCSE grades and tied with the Young et al. estimator for height.

By adding parental PGIs, we estimated the 'direct effects' of PGIs in the EUR sample (Methods and Fig. 6b). The direct effect on GCSE grades of the population-effect educational attainment (EA) PGI was much smaller than the PGI's population effect (Fig. 6a), consistent with previous studies[6,25–27]. In contrast, we did not observe smaller direct

effects than population effects for the DGE PGIs, suggesting that factors not highly correlated with DGEs contribute to the prediction ability of population effect EA PGIs.

PGIs constructed from summary statistics derived in one genetic ancestry tend to predict phenotypes less well in other genetic ancestries[18,28,29]. Although differences in LD patterns and allele frequencies have been argued to be the primary explanation, confounding factors not shared across ancestries could contribute. Therefore, DGE PGIs may predict better across ancestries due to the removal of confounding factors.

We examined cross-ancestry prediction in a sample of 2,214 individuals of predominantly South Asian genetic ancestry (hereafter 'SAS sample'). The population-effect PGIs gave the most accurate predictions in the SAS sample for all phenotypes. However, for height, the best-performing DGE PGI (from the unified estimator) performed nearly as well as the population-effect PGI. As expected[18,28], the prediction accuracy of the population-effect PGIs was lower in the SAS sample than in the EUR sample. In contrast, for height and BMI, the prediction accuracy for the DGE PGIs was higher in the SAS sample than in the EUR sample (Fig. 6c). This difference was statistically significant for the unified estimator prediction on height ($\beta_{SAS} - \beta_{EUR} = 0.085$; standard error (s.e.) = 0.0392; $P = 0.0305$, two-sided Z-test) (Supplementary Table 4).

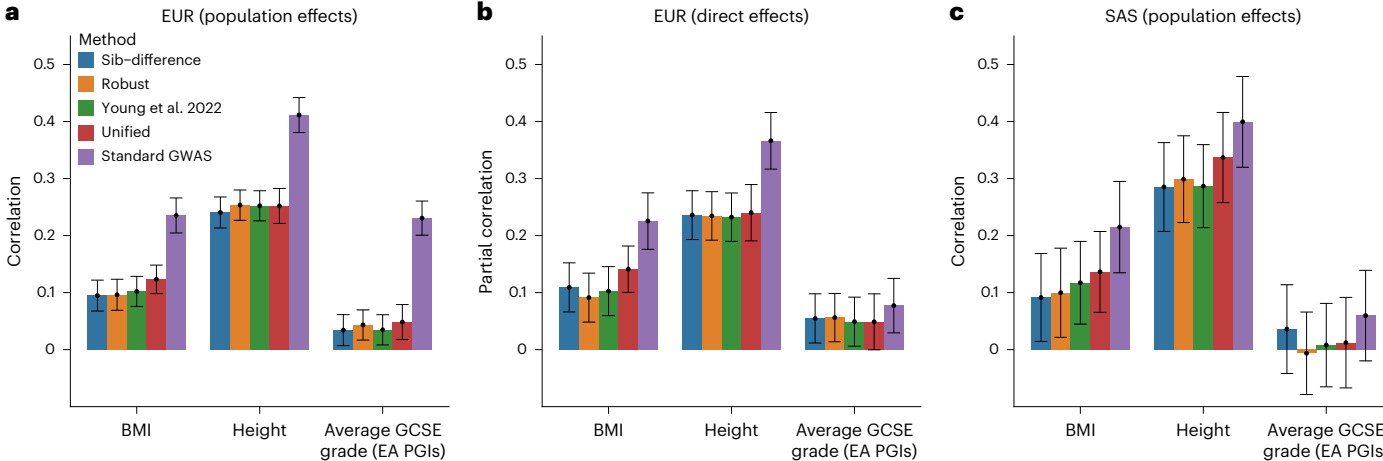

**Fig. 6 | Prediction of phenotypes in the MCS.** We computed PGIs for BMI, height and educational attainment (EA) using summary statistics produced by different estimators applied to UKB data (Methods and Table 2). We use GCSE grades as the outcome for the EA PGI because MCS samples are too young to have completed their education. A GCSE (general certificate of secondary education) is an academic qualification based on exams taken at age 16 by nearly all students in England. The outcome here is the average of a transformation of English and Mathematics GCSE grades to normally distributed Z-scores (Methods). Both the sib-differences and robust estimators were applied to UKB data without ancestry restrictions, whereas the other estimators were applied to the white British ancestry subsample of the UKB. Phenotypes and PGIs were normalized

to have variance 1, so that the 'population effect' of the PGI corresponds to its partial correlation with the phenotype, and the 'direct effect' of the PGI, which is the regression coefficient controlling for parental PGIs (Methods), also corresponds to a partial correlation coefficient. **a**, 'Population effect' of PGIs in European ancestry (EUR) subsample. **b**, 'Direct effect' of PGIs in EUR subsample. **c**, 'Population effect' of PGIs in South Asian (SAS) ancestry subsample. Error bars give 95% confidence intervals. EUR and South Asian ancestry (SAS) subsamples were defined in reference to 1000 Genomes[24] superpopulations (Methods). EUR sample sizes: 5,285 for BMI, 5,285 for height and 4,145 for EA. SAS sample sizes: 685 for BMI and height and 615 for EA. We did not estimate direct effects of the PGIs in the SAS sample due to its small size.

## Discussion

We introduced three family-based estimators of DGEs (Tables 1 and 2): the 'unified estimator' (Fig. 1), which increases the effective sample size for DGEs by inclusion of singletons while producing estimates of population effects equivalent to what would be obtained from standard GWAS in a homogeneous ancestry sample (Extended Data Fig. 1), and two estimators that are robust to population structure and are more powerful than sib-differences, the NT and robust estimators, with the robust estimator also being robust to admixture.

We compared the estimators in a bias-variance framework for simulated populations with different levels of population structure (Fig. 4). From this, we can order the different estimators (Table 2) based on increasing effective sample size (statistical power): sib-difference, robust, NT, Young et al., unified and standard GWAS. This reflects the ordering in terms of bias due to population structure/admixture, except for the sib-difference and robust estimators, which are both robust to population structure and admixture. We recommend the unified estimator for the homogeneous samples ($F_{st} \leq 0.01$) typically used in standard GWAS and the robust estimator for samples with stronger structure ($F_{st} > 0.01$) and/or recent admixture.

We found that standard GWAS with PC adjustment generally did not fully control for stratification when structure was complex (100 subpopulations). A related question is the degree to which stratification confounding affects GWAS of rare variants, which track recent structure in the population that PCs derived from common variants do not capture well[10,30]. Although the estimators studied here could be applied to remove confounding from rare variant analyses, power will be limited at current sample sizes.

We investigated imputing missing parental genotypes using more distant relatives, such as cousins (Methods and Supplementary Note 3). However, we found that imputation from more distant relatives introduces an unacceptable degree of confounding into DGE estimates. We therefore do not recommend imputation from more distant relatives for DGE estimation.

We applied the estimators to 19 phenotypes in the UKB, demonstrating that the unified estimator can give a substantial gain in

effective sample size for DGEs over both the Young et al. estimator (up to 42.6%) and sib-differences (up to 106.5%). We applied the robust estimator to UKB samples without ancestry restrictions, giving effective sample sizes between 10.3% and 21.0% greater than sib-differences. Although not true for the UKB, the robust estimator could be more powerful than the unified estimator for samples that cannot be partitioned into homogeneous ancestry subsamples.

We investigated the performance of polygenic predictors (PGIs) derived from the different estimators in EUR and SAS genetic ancestry samples from the MCS (Fig. 6). The unified estimator generally performed the best out of the PGIs constructed from DGE estimates (DGE PGIs), reflecting its larger effective sample size. We found suggestive evidence that DGE PGIs predict better across ancestries than population-effect PGIs. However, analysis of DGE PGIs in more non-EUR samples is needed before firm conclusions can be drawn.

We have presented a set of estimators that maximize power for estimating DGEs in different scenarios while having no or negligible confounding due to population stratification. We have implemented the estimators in a computationally efficient LMM that accounts for sample relatedness and shared sibling environment, available in the software package snipar ('Code availability'). This will facilitate production of powerful DGE estimates from diverse ancestries that can be used in downstream applications including estimation of heritability and genetic correlations[13–15], inference of natural selection[9,16,19] and Mendelian randomization[17].

## Online content

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

## Methods

Our research conforms with all relevant ethical regulations. UKB has approval from the North West Multi-centre Research Ethics Committee as a Research Tissue Bank approval. This approval means that researchers do not require separate ethical clearance and can operate under the Research Tissue Bank approval (there are certain exceptions to this which are set out in the Access Procedures, such as recontact applications). The MCS has obtained ethical approval from NHS Research Ethics Committees.

### FGWAS designs

FGWAS is defined by regression of phenotype onto genotype and parental genotype(s)[3]:

$$y_{ij} = \delta g_{ij} + \alpha g_{\mathrm{par}(i)} + \epsilon_{ij} \qquad \text{(Model 1)}$$

where $y_{ij}$ is the phenotype of the sibling $j$ in family $i$; $g_{ij}$ is the corresponding genotype; $g_{\mathrm{par}(i)} = g_{p(i)} + g_{m(i)}$ is the sum of paternal and maternal genotypes; $\delta$ is the DGE; $\alpha$ is the average NTC; and $\epsilon_{ij}$ is the residual. Because $\mathbb{E}[g_{ij}|g_{p(i)}, g_{m(i)}] = g_{\mathrm{par}(i)}/2$, and variation in offspring genotype around this expectation is due to random Mendelian segregations (where chromosomes segregate independently of each other and environment), estimates of DGEs from fitting Model 1 are free from confounding due to gene-environment correlation (including population stratification) and correlations with genetic variants on other chromosomes due to nonrandom mating (including AM)[3–5,32]. The average NTC—so named because it equals the average of the coefficients on parental alleles not transmitted to the offspring in a regression of offspring phenotype onto transmitted and NT alleles—captures IGEs from relatives and confounding due to gene-environment correlation and nonrandom mating[3–5,25]. (The FGWAS regression equation here only applies to the autosome. Although it would be possible to apply a similar approach to maternally inherited X chromosomes, FGWAS analyses of other sex chromosomes or mitochondria are not possible.)

Alternatively, one can fit a model that allows for different coefficients on the paternal and maternal genotypes:

$$y_{ij} = \delta g_{ij} + \alpha_p g_{p(i)} + \alpha_m g_{m(i)} + \epsilon_{ij} \qquad \text{(Model 2)}$$

where $\alpha_p$ and $\alpha_m$ are, respectively, the paternal and maternal NTCs. Model 1 can be derived from Model 2 (with a change of residuals[3]), implying that $\alpha = (\alpha_p + \alpha_m)/2$. Although Model 1 is sufficient to remove confounding from estimates of DGEs, irrespective of whether $\alpha_p = \alpha_m$, Model 2 may be preferred or required in certain contexts (Supplementary Note 2.2).

Standard GWAS performs a regression of phenotype onto genotype, giving an estimate of the population effect, $\beta$. Assuming random mating, it can be shown that[3]: $\beta = \delta + \alpha$. This provides a useful connection between the parameters of FGWAS and standard GWAS.

Fitting Models 1 and 2 entails restricting one's sample to those with both parents genotyped, which is often only a small fraction (or none) of the sample. Genetic differences between siblings, which are randomly assigned, can be used instead[12,16]. For example, one can perform the following regression:

$$y_{i1} - y_{i2} = \delta(g_{i1} - g_{i2}) + \epsilon_{i1} - \epsilon_{i2}. \qquad \text{(Model 3)}$$

Estimates of $\delta$ from this model, which we call 'sib-differences', are free from confounding due to nonrandom mating and most gene-environment correlation, the exception being (unlike estimates from Models 1 and 2) confounding due to IGEs from siblings[3]. In addition, estimates of DGEs from Model 3 are less precise than those from Model 1 or 2 when applied to sibling data provided that the correlation between siblings' residuals is modeled, as in a generalized

least-squares estimator[3]. Furthermore, estimation of Model 3 ignores samples with genotyped parent(s) but without genotyped siblings[16].

### Imputing missing parental genotypes for singletons

An alternative approach to estimating DGEs using sibling data was proposed by Young et al.[3]: treat parental genotypes as missing data and impute them according to Mendelian laws. For a sibling pair, the missing parental genotype, $g_{\mathrm{par}(i)}$, is imputed conditional on the identity-by-descent (IBD) state of the siblings; that is, whether the siblings have inherited the same or different alleles from each parent. Young et al. developed this approach, termed 'Mendelian imputation', for all samples with at least one genotyped first-degree relative, not just sibling pairs. The resulting imputed parental genotypes are then used in place of the observed ones in Model 1 or 2.

Here, we extend the Mendelian imputation approach to singletons, samples without a genotyped first-degree relative. We observe two out of four parental alleles in a singleton—the same as for a sibling pair in IBD2, meaning they have inherited the same alleles from both the mother and father[3]. Under random mating, the imputed parental genotypes are:

$$\hat{g}_{\mathrm{par}(i)} = \mathbb{E}[g_{\mathrm{par}(i)}|g_i] = g_i + 2f; \hat{g}_{p(i)} = \mathbb{E}[g_{p(i)}|g_i] = g_i/2 + f = \hat{g}_{m(i)} \qquad (1)$$

where the two unobserved alleles are imputed using the allele frequency, $f$. If the imputed parental genotypes are unbiased, then the DGE estimates obtained when including singletons in Models 1 and 2 will be unbiased and consistent, provided that the resulting regression design matrix is not collinear[3]. As the imputation from a singleton is linear, singleton data alone cannot be used to identify DGEs, because the design matrix would be collinear. Genotype-phenotype data from individuals with genotyped first-degree relatives, where a non-linear imputation of parental genotype(s) is possible[3], is needed in addition to singletons.

### Imputing parental genotypes using higher-degree relatives

The linear imputation (equation (1)) used for singletons in the unified estimator ignores information on parental genotypes from more distant relatives than siblings and parents, such as aunts/uncles and cousins. We investigated whether imputing missing parental genotypes using higher-degree relatives could improve estimation of DGEs.

We simulated cousin pairs and performed imputation of missing parental genotypes using the cousin pair's genotypes (Supplementary Note 3). The imputed parental genotypes were approximately unbiased and more accurate than when imputing from a single offspring (as in the unified estimator). However, DGE estimates when using parental genotypes imputed from cousins showed substantial population stratification confounding even for relatively homogeneous ancestry samples ($F_{st} \geq 0.001$) (Extended Data Fig. 5).

The reason that higher-degree relatives introduce bias is that they are separated by more than one meiosis. This implies that the genotype of the relative is not randomly assigned—and therefore not independent of confounds—conditional on the missing parental genotype. For example, your cousin's genotype is not randomly assigned conditional on your parent's genotype. This implies that the genotype of a cousin contains information not only on your parent's genotype but also information that can reflect confounds such as population structure.

### Population structure robust estimators

Young et al.[3] proposed an alternative, imputation-based estimator for sibling-pair data that they argued should not be biased by population structure. This estimator partitioned the sibling pairs based on their IBD state and performed separate regressions for sibling pairs in IBD0 (no alleles shared by descent from parents) and IBD1 (one allele shared by descent from parents) followed by an inverse-variance-weighted

meta-analysis of the DGE estimates. Young et al.[3] showed this is more powerful than sib-differences, giving an effective sample size $1 + \frac{1-r}{6(1+r)}$ times greater, where $r$ is the correlation of siblings' residuals. However, this estimator has a smaller effective sample size than the primary estimator considered in Young et al.[3], which includes sibling pairs in IBD2, at the cost of potential bias due to population structure.

The NT estimator we develop is a generalization of this estimator that partitions the sample based upon which NT parental alleles have been observed (Table 1). Although the NT estimator is robust to an island model of population structure and more powerful than sib-differences even for sibling pair data (Supplementary Note 2.3.2), we found that it is biased when paternal and maternal allele frequencies differ, as in recently admixed samples (Supplementary Note 2.3.3).

The robust estimator is similar to the NT estimator, except that it performs different regressions in three of the four groups (Table 1). It performs uniparental regressions for the samples with one parental NT allele observed when the parent-of-origin is known. For a parent-offspring pair that are both heterozygous, phased data are required to determine the parent-of-origin[3]. For sibling pairs in IBD1 without genotyped parents, the parent-of-origin of alleles is unknown, and we perform a regression controlling for the mean genotype in the sibship, equivalent to sib-differences (Supplementary Note 2.5). However, for sibling pairs in IBD1 with a genotyped parent, we partition the sample based on whether the allele shared by the siblings is from the observed parent or the missing parent: when the shared allele is from the missing parent, we perform uniparental regressions using the observed parental genotype and the alleles inherited by the siblings from that parent (for example, a sibling pair in IBD1 with a genotyped mother would be placed in the maternal NT group when the shared allele is from the father); but when the shared allele is from the observed parent, we are able to fully recover the missing parent's alleles and place the siblings in the both NT group. (See Supplementary Note 5.2 from Young et al.[3] for further details on determining shared alleles for cases with one parent and multiple full-sibling offspring with observed genotypes.)

Thus, to implement the robust estimator when some samples have one but not both parents' genotypes observed, the imputation procedure in snipar with phased data should be performed first. This will determine how many NT parental alleles have been observed for each sample with a genotyped first-degree relative and the parent-of-origin of alleles for the samples with one parent genotyped.

## Linear mixed model inference

Here, we develop an LMM that generalizes the LMM used in Young et al.[3] and the LMM implemented in fastGWA[33], which is specified by a sparse GRM. This approach ensures that residual correlations between siblings are modeled properly, ensuring statistically efficient estimates of DGEs are obtained while also modeling residual correlations between all pairs related above some threshold, thereby ensuring statistically efficient estimates with accurate standard errors are obtained when more complex relatedness is present in the sample[33].

Stacking all observation vertically, for a dataset with $N$ individuals in $n$ families, the model is

$$\mathbf{y} = X\boldsymbol{\theta} + \mathbf{e}'$$

where $\mathbf{y}$ is the $N \times 1$ phenotype vector; $X$ is the $N \times c$ matrix specifying the fixed effects, where the columns of $X$ depend upon the covariates and estimator being used (Supplementary Note 2); $\boldsymbol{\theta}$ is the corresponding vector of fixed effects; and $\mathbf{e}'$ is a random vector, which we specify below. For example, if fitting Model 2 without additional covariates, $\boldsymbol{\theta} = [\delta, \alpha_p, \alpha_m]^\top$, and $X$ has columns giving proband, (imputed or observed) paternal, and (imputed or observed) maternal genotypes. The random vector $\mathbf{e}'$ is specified as

$$\mathbf{e}' = \mathbf{g} + Z\mathbf{u} + \mathbf{e}$$

where

$$\mathbf{g} \sim \mathcal{N}\left(\mathbf{0}, \sigma_g^2 \Pi\right);$$

$\Pi$ is the (sparse) GRM; $\sigma_g^2$ is the corresponding variance parameter; $Z$ is an $N \times n$ sibship indicator matrix, with entry $k, l$ equal to 1 if the $k$th individual is in sibship $l$ and 0 otherwise; and $\mathbf{u}$ is an $n \times 1$ normally distributed sibship-specific mean vector

$$\mathbf{u} \sim \mathcal{N}\left(\mathbf{0}, \sigma_s^2 I_n\right),$$

where $\sigma_s^2$ is the sibship covariance parameter. The sibship covariance component $Z\mathbf{u}$ is thus also normally distributed:

$$Z\mathbf{u} \sim \mathcal{N}\left(0, \sigma_s^2 Z Z^\top\right).$$

The residual variance vector has distribution:

$$\mathbf{e} \sim \mathcal{N}\left(\mathbf{0}, \sigma_\epsilon^2 I_N\right).$$

Therefore, the variance-covariance matrix of $\mathbf{y}|X$ is $V = \sigma_g^2 \Pi + \sigma_s^2 Z Z^\top + \sigma_\epsilon^2 I_n$.

The relatedness matrix, $\Pi$, can be either an SNP-based GRM or a GRM computed from IBD segments, such as output by KING[34]. By setting elements of $\Pi$ below a certain threshold, usually 0.05, to zero, the sparsity of the $V$ matrix can be exploited so that restricted maximum likelihood (REML) inference of variance components and the generalized least-squares estimate of $\theta$ given the variance components are computationally feasible even for large-scale biobanks[33] (Supplementary Table 2). For analyses in this paper, we used a relatedness matrix constructed from KING IBD segments with a 0.05 threshold. We chose this threshold as it enables accurate modeling of residual correlations between close relatives[33,35] without requiring prohibitive memory usage and computation time. However, users can specify a different threshold to the software.

## Variance component estimation

The variance component parameters $\sigma_g^2, \sigma_s^2, \sigma_\epsilon^2$ are estimated by maximizing the REML log likelihood function:

$$L = -(\log|V| + \log|C^\top V^{-1} C| + \mathbf{y}^\top P\mathbf{y})/2,$$

where $C$ is the design matrix of fixed covariates, and

$$P = V^{-1} - V^{-1}C\left(C^\top V^{-1} C\right)^{-1} C^\top V^{-1}.$$

If no fixed covariates are included, $C$ is a column vector of all 1s.

If the relatedness matrix $\Pi$ is dense, then $V$ is dense, leading to resource-demanding computation. To reduce the computational burden, we follow Jiang et al.[33] and zero out entries in $\Pi$ with relatedness below a default threshold of 0.05. This results in a highly sparse matrix, enabling the use of efficient sparse matrix algorithms for likelihood evaluation. By using a gradient-free optimizer, REML variance component estimation can be done in just a few minutes for datasets as large as the UKB. Another possible benefit is that, by considering only close relatives, the correlations between close relatives are modeled more accurately than when using a SNP-based relatedness matrix that includes relatedness measures between all pairs[15,33].

With a sparse $V$, we compute $V^{-1}\mathbf{y}$ and $V^{-1}C$ using a sparse LU solver in SciPy (v1.7.2)[36], without explicitly computing $V^{-1}$. Then variance component parameters are optimized using the gradient-free L-BFGS algorithm[36]. One can choose to model only the sibship variance component and the residual variance component as in Young et al.[3], which also results in a sparse $V$ matrix, so the same computational procedure can be used in this case.

## Estimating SNP effects

To include covariates in the genome-wide estimation of SNP effects, we project both genotypes and phenotypes onto the space orthogonal to the space spanned by the covariates, as in BOLT-LMM[37]:

$$\bar{\mathbf{X}} = M_c X \text{ and } \bar{\mathbf{y}} = M_c \mathbf{y}$$

where $M_c = I_N - C(C^\top C)^{-1} C^\top$ is the projection matrix. Then the effect estimates are given by

$$\hat{\theta} = \left(\bar{X}^\top V^{-1} \bar{X}\right)^{-1} \bar{X}^\top V^{-1} \bar{\mathbf{y}},$$

where

$$\mathrm{Var}(\hat{\theta}) = \left(\bar{X}^\top V^{-1} \bar{X}\right)^{-1}$$

is the sampling variance-covariance. By the Frisch-Waugh-Lovell Theorem, this gives estimates of the SNP effects that are equivalent to performing the joint-regression on the covariates and the proband and relative genotype(s).

The procedure for the NT and robust estimators is more complicated, as we need to account for covariance across the estimates of DGEs from the different groups (Table 1) due to relatedness across the groups. We describe the procedure in Supplementary Note 2.4.1.

## Simulations of structured populations

For different levels of $F_{st}$, we generated $K$ subpopulations. We simulated SNPs from binomial distributions, where subpopulation allele frequencies were drawn from the Balding-Nichols model[31]: $f_k \sim \mathrm{Beta}(\frac{1-F_{st}}{F_{st}} f, \frac{1-F_{st}}{F_{st}}(1-f))$, where $f_k$ is the allele frequency in subpopulation $k$ and $f = E_k[f_k]$ is the overall allele frequency.

We investigated two different scenarios: a simple scenario with two subpopulations ($K = 2$), where each subpopulation has 20,000 families each with two siblings, and all overall allele frequencies are 0.5 ($f = 0.5$); and a more complex scenario with 100 subpopulations ($K = 100$), each with 1,000 families, and overall allele frequencies drawn from a distribution proportional to $1/f$ for $0.5 > f > 0.05$. We chose this distribution as it reflects the distribution of allele frequencies for a randomly mating population with constant effective size[38]. We simulated 20,000 SNPs for the first scenario and 4,000 SNPs for the second scenario.

We generated phenotypes with 50% of the phenotypic variance attributed to subpopulation phenotype means $\mu_k$ that we sampled independently from a mean-zero normal distribution: $\mu_k \sim N(0, \sigma_\mu^2)$. The remaining 50% of the phenotypic variance was attributed to random Gaussian noise, implying a correlation between siblings' phenotypes of 0.5. There are no causal effects (including DGEs) of the genotypes in this simulation, so any deviation from the null distribution is evidence of bias due to population stratification.

The average confounding bias due to population stratification in this model is zero, but the average magnitude of the confounding is non-zero for a finite number of subpopulations $K$. To see this, consider the overall covariance between the genotype $g$ at an SNP $l$ and phenotype $Y$, which is the covariance between the subpopulation genotype means ($2f_k$) and subpopulation phenotype means ($\mu_k$):

$$\mathrm{Cov}(g, Y) \approx \sum_k p_k 2(f_k - f)\mu_k,$$

where $f_k$ is the allele frequency in subpopulation $k$, $f = \sum_k p_k f_k$ is the overall allele frequency, and $p_k$ is the fraction of families in subpopulation $k$. The regression coefficient of genotype onto phenotype is therefore

$$\beta_f = \frac{\mathrm{Cov}(g, Y)}{\mathrm{Var}(g)} \approx \frac{\sum_k p_k 2(f_k - f)\mu_k}{\mathrm{Var}(g)}.$$

Because the allele frequencies and subpopulation phenotype means are sampled independently, the regression of phenotype onto genotype has expectation zero across the SNPs but varies around zero due to the finite number of subpopulations. We quantify this through the expected squared regression coefficient across SNPs with overall allele frequency $f$:

$$E[\beta_f^2] \approx \frac{\sum_k 4p_k^2 F_{st} f(1-f) \sigma_\mu^2}{\mathrm{Var}(g)^2},$$

where we have used the variance of the allele frequencies from the Balding-Nichols model ($F_{st} f(1-f)$) to obtain this. If we assume equal subpopulation sizes, then $p_k = 1/K$ and

$$E\left[\beta_f^2\right] \approx \frac{4 F_{st} f(1-f) \sigma_\mu^2}{K \mathrm{Var}(g)^2} = \frac{F_{st} \sigma_\mu^2}{K f(1-f)(1 + F_{st})^2},$$

where we have used the fact[3] that $\mathrm{Var}(g) = 2f(1-f)[1 + F_{st}]$. The expected magnitude of the bias therefore decreases (towards zero) with the number of subpopulations, holding $f$, $F_{st}$, and $\sigma_\mu^2$ constant. If we consider the expected phenotypic variance explained (in a regression sense, that is the $R^2$) by each variant, it does not depend upon overall allele frequency:

$$\frac{\mathrm{Var}(g) E\left[\beta_f^2\right]}{\mathrm{Var}(Y)} \approx \frac{2 F_{st} \sigma_\mu^2}{K(1 + F_{st})\mathrm{Var}(Y)}.$$

For the simulations involving the unified estimator, we sought to mimic the fact that large biobanks such as the UKB consist mostly of singletons. For 90% of families, we randomly removed one sibling to obtain singletons, leaving 10% of families with two siblings. The sibling-difference/robust, NT, and Young et al. estimators were applied to the 10% of families with intact sibling pairs, whereas the unified estimator and standard univariate GWAS were applied to the combined sample of singletons and sibling pairs.

We also examine the performance of the estimators in a sibling-pair-only scenario: that is, 20,000 genotyped and phenotyped sibling pairs in each subpopulation (Extended Data Fig. 4). We applied the estimators to the resulting 40,000 sibling pairs. Note that in this scenario, there are no singletons, and the unified estimator is equivalent to the Young et al. estimator.

To assess evidence for bias due to population stratification, we computed the mean of the squared Z-scores, that is, $\hat{\delta}^2/\mathrm{Var}(\hat{\delta})$ (or $\hat{\beta}^2/\mathrm{Var}(\hat{\beta})$ for standard GWAS estimates of 'population effects'), of the estimated SNP effects produced by different estimators, which should be 1 in expectation under the null and will be above 1 in expectation if there is bias due to population stratification. (See above for a derivation of the expected squared bias for standard GWAS.) Although a mean $Z^2$ statistic greater than 1 is a common measure of inflation in GWAS[39,40], this statistic is not a completely fair way to compare the biases due to stratification across estimators that have different sampling variances: for example, for estimators with the same bias but different sampling variances, the estimator with the smaller sampling variance would be expected to produce larger $Z^2$ statistics on average. For this reason, we also examine the nonsampling variance of an estimator $\zeta$, $B_\zeta^2$, across all $L$ SNPs ($L = 20,000$ for first scenario and $L = 4,000$ for the second scenario), which we estimate as

$$\hat{B}_\zeta^2 = \frac{1}{L}\sum_{i=1}^{L} \hat{\zeta}_i^2 - \frac{1}{L}\sum_{i=1}^{L} \mathrm{Var}(\hat{\zeta}_i)(\zeta = \delta, \text{ or } \beta \text{ for standard GWAS}). \qquad (2)$$

Denoting by $b_{\zeta i}^2$ the expected squared bias of an estimator $\zeta$ at an SNP $i$, the expectation of the estimator's nonsampling variance is

$$\mathbb{E}[\hat{B}_\zeta^2] = \frac{1}{L}\sum_{i=1}^{L} b_{\zeta i}^2.$$

Thus, $\hat{\beta}_\zeta^2$ is an estimate of the magnitude of bias due to population stratification that can be fairly compared across estimators.

## Analysis of UKB data

We selected 19 UKB phenotypes related to education, cognition, income and health. Phenotypes were derived from baseline measurements. Note that 'cognitive ability' is derived from Field 20016 ('Fluid intelligence score')[20]. Each phenotype was normalized to have mean 0 and variance 1 within each sex. More details can be found in Okbay et al.[25].

We filtered out samples that had been flagged as having the following QC issues: excess relatives, sex chromosome aneuploidy, and/or identified as outliers in heterozygosity or genotype missingness. We used the phased haplotypes for the UKB genotyping array SNPs provided as part of the UKB data release. We filtered out variants with minor allele frequencies less than 0.01 and with Hardy-Weinberg equilibrium exact test $P$ value less than $1 \times 10^{-6}$, resulting in 658,720 SNPs. We inferred IBD segments shared between siblings and performed imputation using snipar[41].

We implemented the estimators in the LMM described above[41]. We inferred sibling relationships using the KING (v2.2.5) software[34] with the '−related −degree 1' argument. The sparse GRM is derived from IBD segments inferred by KING with the argument '−ibdseg −degree 3', with the relatedness threshold set at 0.05. For the Young et al. and unified estimators, we fit Model 2 for each SNP, substituting imputed parental genotypes for observed parental genotypes when not available. Details of the implementation of the sib-difference and robust estimators are in Supplementary Notes 2.4 and 2.5. We derived standard GWAS 'population effect' estimates as the sum of the DGE and average NTC estimates from the unified estimator. We adjusted for age and 40 genetic PCs, and, for phenotypes not specific to one sex, we also adjusted for sex, the interaction between sex and age, and the interaction between sex and age up to the third order.

To compute the relative effective sample sizes of the different estimators, we analyzed 10,911 SNPs on chromosome 22 (Fig. 5 and Supplementary Table 1). For the Young et al., unified, robust, and sib-difference estimators, we computed genome-wide summary statistics for height, EA, and BMI, and summary statistics for SNPs on chromosome 22 for all other phenotypes ('Data availability').

We computed genetic correlations between DGE estimates from the four methods for height and EA, respectively, using LDSC (v1.0.1)[39,42]. All genetic correlation estimates are close to 1 (Supplementary Table 3): for example, $r_g$ between DGE estimates from the Young et al. and the robust estimator is 1.0034 (s.e. = 0.0053) for height and 0.9925 (s.e. = 0.0265) for EA.

## Polygenic prediction in the MCS

We chose the MCS[43] as our validation sample because it is a nationally representative sample (of 8,202 individuals born around the year 2000 in the United Kingdom) and the cohort members are too young to have participated in the UKB, ruling out sample overlap (although some older relatives of MCS cohort members could be present in the UKB). Furthermore, both parents' genotypes are available for 3,421 cohort members, and one parent's genotype is available for 3,989 cohort members.

We projected MCS samples onto the top 20 PCs derived from 1000 Genomes data[24] using the OADP algorithm[44] implemented in bigsnpr (v1.12.2)[45], and we determined the European and South Asian ancestry subsamples as those with 20 nearest neighbors all in 1000 Genomes EUR and SAS superpopulations, respectively (Extended Data Fig. 6 shows a visualization of the sample PC distribution). We used PRS-CS (4 June 2021)[46] and the provided UKB European LD panel to obtain posterior SNP weights for PGIs for BMI, height and EA.

We evaluated the performance on three widely studied quantitative phenotypes: BMI, height and educational achievement.

Educational achievement is measured by mathematics and English GCSE grades, which are exams taken at age 16 by nearly all students in England (Methods and Supplementary Note 4). The MCS phenotypes were derived from sweep 7, which was performed when cohort members were aged 17 years. The validation phenotypes were standardized to have mean zero and variance one within each sex. We used height and BMI measured at age 17. The educational achievement outcome was derived from the average of English and Mathematics GCSE grades transformed into Z-scores (Supplementary Note 4).

To estimate the 'population effect' of the PGI, we performed the following regression separately in EUR and SAS samples:

$$Y_i = \alpha_0 + \beta_{\mathrm{PGI}}\mathrm{PGI}_i + X\mathbf{b} + \epsilon_i$$

where $Y_i$ is the phenotype observation for genotyped individual $i$; $\mathrm{PGI}_i$ is the PGI value; $\alpha_0$ is the intercept; $\beta_{\mathrm{PGI}}$ is the population effect of the PGIs; $X$ is the design matrix of the first 20 PCs; $\mathbf{b}$ is the vector of regression coefficients for the PCs; and $\epsilon_i$ is the residual. The population effect here is equivalent to a partial correlation coefficient because both phenotype and PGI have been scaled to have variance 1.

To estimate the 'direct effect' of the PGI[25], we performed the following regression in the EUR sample:

$$Y_i = \alpha_0 + \delta_{\mathrm{PGI}}\mathrm{PGI}_i + \alpha_{\mathrm{PGI}:p}\mathrm{PGI}_{p(i)} + \alpha_{\mathrm{PGI}:m}\mathrm{PGI}_{m(i)} + X\mathbf{b} + \epsilon_i,$$

where $\mathrm{PGI}_{p(i)}$ and $\mathrm{PGI}_{m(i)}$ are, respectively, the paternal and maternal PGIs (constructed using the same weights as the proband PGI); and $\alpha_{\mathrm{PGI}:p}$ and $\alpha_{\mathrm{PGI}:m}$ are, respectively, the paternal and maternal NT coefficients of the PGI[3]. When a parent's genotype was missing, the parent's PGI was computed from imputed parental genotypes, as in Young et al.[3].

## Reporting summary

Further information on research design is available in the Nature Portfolio Reporting Summary linked to this article.

## Data availability

Summary statistics from the different estimators applied to UKB data are available for download from the SSGAC data portal: https://thessgac.com/. Applications for access to the UKB data can be made on the UKB website (http://www.ukbiobank.ac.uk/register-apply/). Applications for MCS data can be made by following the instructions here: https://cls.ucl.ac.uk/data-access-training/data-access/accessing-data-directly-from-cls/. 1000 Genomes phase 3 data can be downloaded using the download_1000G function provided by the bigsnpr[45] R package.

## Code availability

The sibling and family-based GWAS estimators investigated here are implemented in the software package snipar: github.com/AlexTISYoung/snipar/. The specific code used for the results reported in this paper is available here[41]: https://github.com/AlexTISYoung/snipar/releases/tag/v0.0.19.

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

## Acknowledgements

The study was supported by Open Philanthropy and the National Institute on Aging/National Institutes of Health through grants R24-AG065184 and R01-AG042568 to D.J.B., R01-AG083379 to A.S.Y. and R01-AG081518 to P.T. This research has been conducted using the UKB Resource under Application Number 11425.

## Author contributions

A.S.Y. conceived the study. J.G. derived theoretical results. A.S.Y. and J.G. designed the simulations. J.G. performed the simulations. S.M.N. wrote the imputation code. J.G. analyzed the UKB data. J.G., T.T. and M.B. analyzed the MCS data. A.S.Y., J.G., P.T. and D.J.B. wrote the paper.

## Competing interests

The authors declare no competing interests.

## Additional information

**Extended data** is available for this paper at https://doi.org/10.1038/s41588-025-02118-0.

**Correspondence and requests for materials** should be addressed to Junming Guan or Alexander Strudwick Young.

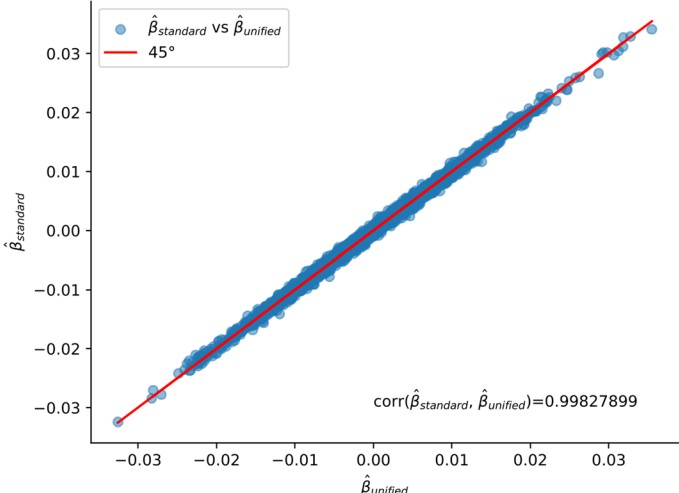

**Extended Data Fig. 1 | Population effect estimates from standard GWAS and the unified estimator for BMI in UKB.** We estimated population effects on BMI for SNPs on chromosome 22 in UKB using both standard GWAS and the unified estimator: $\hat{\beta}_{standard}$'s were estimated from the standard GWAS and $\hat{\beta}_{unified}$'s were calculated as the sum of the direct effect and average parental NTC estimates. The correlation of the two sets of estimates is 0.998. The 45° line (red solid line) represents $\hat{\beta}_{standard} = \hat{\beta}_{unified}$.

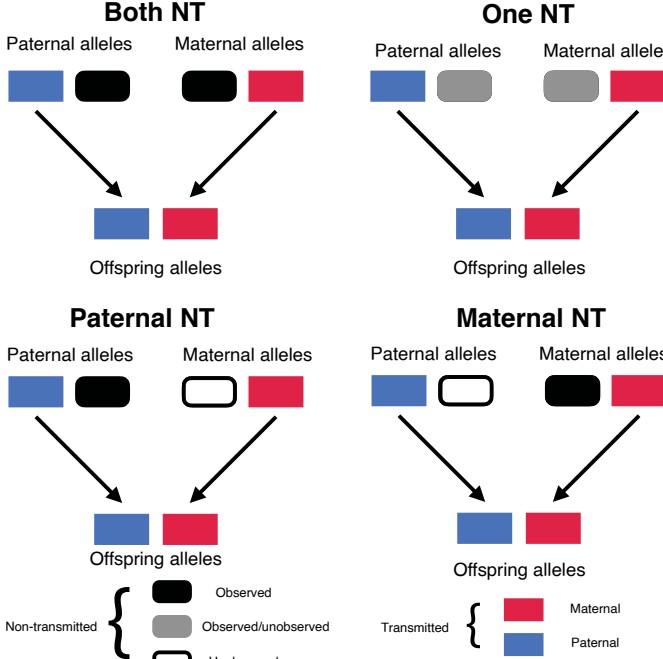

**Extended Data Fig. 2 | Disjoint groups analyzed by the non-transmitted estimator and the robust estimator.** The non-transmitted estimator and the robust estimator partition individuals with at least one non-transmitted (NT) parental allele observed into four disjoint groups (Table 1). The 'Both NT' group is for individuals with both NT parental alleles observed, such as for sibling pairs in IBD0 and parent-offspring trios. The 'One NT' group is for siblings in IBD1 without any genotyped parents, where we have observed one NT parental allele, and it is the maternal or parental allele with equal probability. The 'Paternal NT' group is for individuals for whom we have observed the paternal NT allele but not the maternal, for example, genotyped father-child pairs. The 'Maternal NT' group is for individuals for whom we have observed the maternal NT allele but not the paternal NT allele, for example, genotyped mother-child pairs.

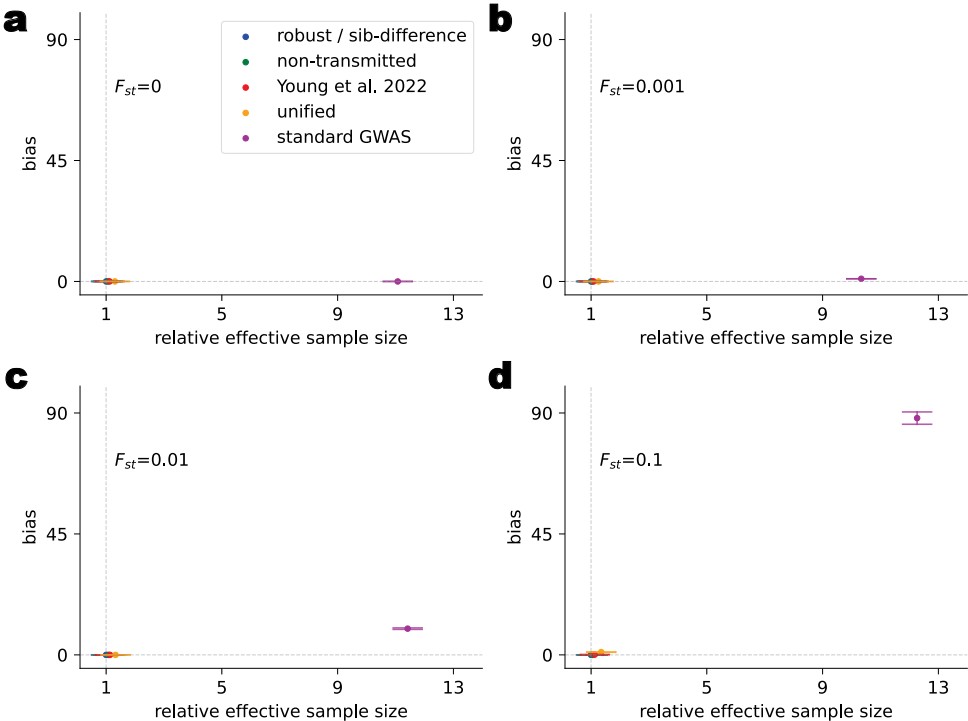

**Extended Data Fig. 3 | Bias-variance tradeoff on simulated sibling pairs and singletons.** See Figs. 2 and 4 in the main text for details on simulation setup. The effective sample size (x-axis) is defined relative to that of the sib-difference or robust estimator (Table 2 in the main text) and should be equal to 1 (vertical dashed line) for the sib-difference or robust estimator and higher than 1 for the other estimators. Bias (y-axis) is measured as the nonsampling variance (Methods section in the main text) relative to that for standard GWAS with $F_{st}$ = 0.001, and is expected to be above 0 (horizontal dashed line) when there is bias due to population stratification. Bias is presented as the mean with a 95% jackknife confidence interval over 20,000 SNPs. (**a**)-(**d**) bias-variance tradeoff comparison for the sibling difference method, robust estimator, Young et al., unified estimator, and standard GWAS with different levels of population structure, as measured by $F_{st}$.

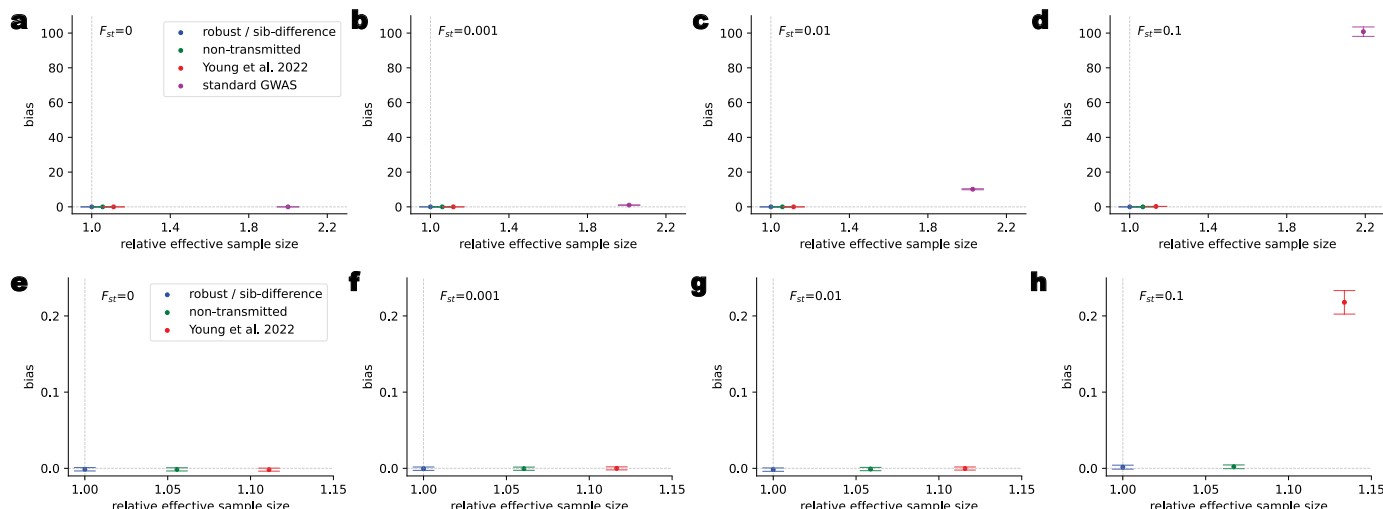

**Extended Data Fig. 4 | Bias-variance tradeoff on simulated sibling pairs.** We simulated populations and phenotypes as in Figs. 2 and 4 in the main text and Supplementary Fig. 3, except that we simulated 40,000 sibling pairs (20,000 in each subpopulation). In this case, the unified estimator and the Young et al. estimator are equivalent, because there are no singletons. The effective sample size (x-axis) is defined relative to that of the sib-difference or robust estimator (Table 2 in the main text) and should be equal to 1 (vertical dashed line) for the sib-difference or robust estimator and higher than 1 for the other estimators.

Bias (y-axis) is measured as the nonsampling variance (Methods section in the main text) relative to that for standard GWAS with $F_{st}$ = 0.001, and is expected to be above 0 (horizontal dashed line) when there is bias due to population stratification. Bias is presented as the mean with a 95% jackknife confidence interval over 20,000 SNPs. (**a**)-(**d**) bias-variance tradeoff comparison for the sibling difference method, robust estimator, Young et al. estimator, and standard GWAS with different levels of population structure, as measured by $F_{st}$; (**e**)-(**h**) the same as (**a**)-(**d**) but with the standard GWAS removed for scale.

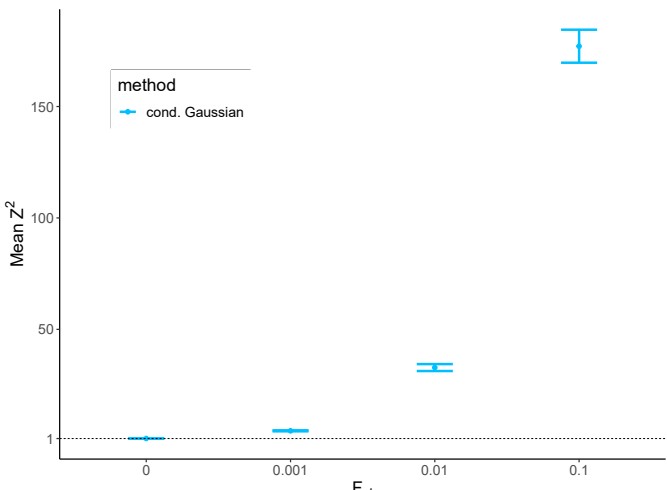

**Extended Data Fig. 5 | Mean $Z^2$ statistic for direct genetic effects when imputing parental genotypes from cousins using the conditional Gaussian formula (Supplementary Note 3).** Mean $Z^2$ statistics, which are expected to be above 1 (dashed line) when there is bias due to population stratification, and the corresponding 95% jackknife confidence intervals over 3,000 SNPs are presented. For each $F_{st}$, we simulated 2 subpopulations, each with 2,500 unrelated cousin pairs and 3,000 SNPs, where the subpopulation allele frequencies were drawn from the Balding-Nichols model: $Beta(\frac{1-F_{st}}{2F_{st}}, \frac{1-F_{st}}{2F_{st}})$; imputation of parental genotypes was carried out using the conditional Gaussian method.

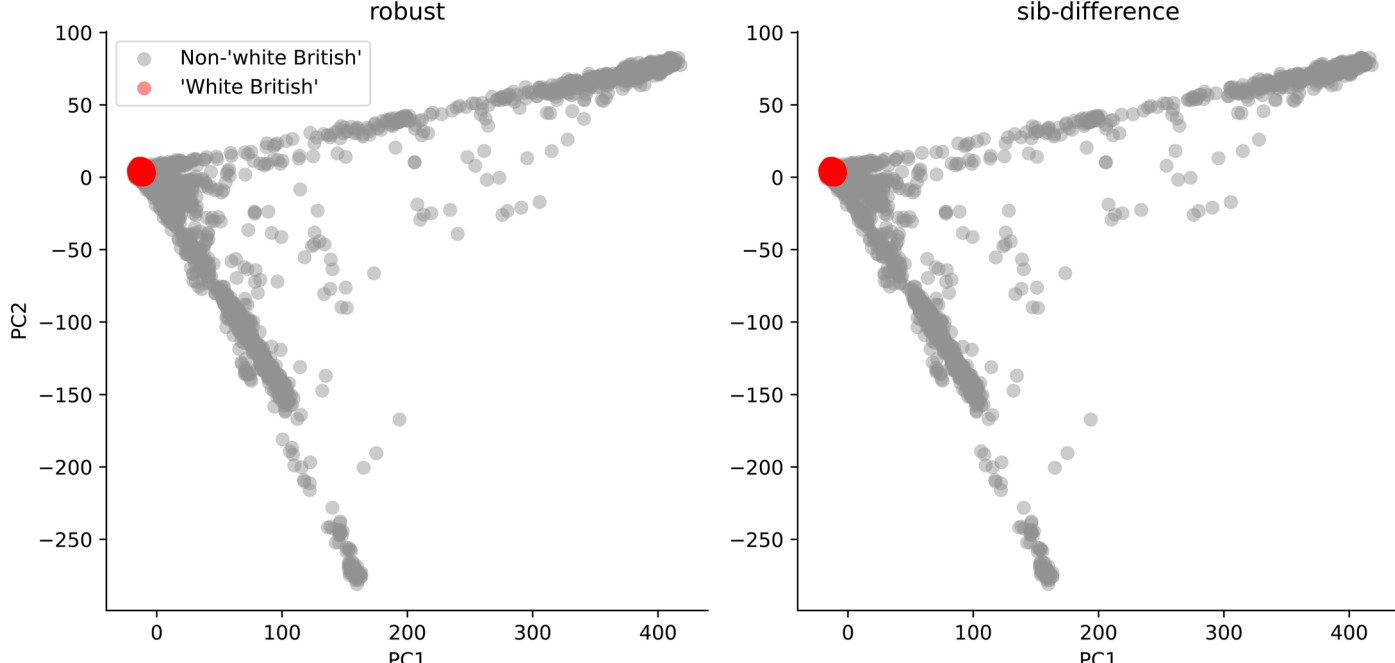

**Extended Data Fig. 6 | The first and second principal components (PC1 and PC2) of UKB samples with at least one genotyped first-degree relative.** There are 51,875 individuals with at least one genotyped relative in UKB, which can be analyzed using the robust estimator; 11.83% (6,135) are non-'White British' and 88.17% (45,740) are 'White British'. 46,698 individuals have at least one genotyped sibling and can be analyzed using the sib-difference estimator; 11.84% (5,528) are non-'White British' and 88.16% (41,170) are 'White British'.

# Reporting Summary

## Statistics

For all statistical analyses, confirm that the following items are present in the figure legend, table legend, main text, or Methods section.

| n/a | Confirmed | |
|---|---|---|
| ☐ | ☒ | The exact sample size (*n*) for each experimental group/condition, given as a discrete number and unit of measurement |
| ☒ | ☐ | A statement on whether measurements were taken from distinct samples or whether the same sample was measured repeatedly |
| ☐ | ☒ | The statistical test(s) used AND whether they are one- or two-sided<br>*Only common tests should be described solely by name; describe more complex techniques in the Methods section.* |
| ☐ | ☒ | A description of all covariates tested |
| ☐ | ☒ | A description of any assumptions or corrections, such as tests of normality and adjustment for multiple comparisons |
| ☐ | ☒ | A full description of the statistical parameters including central tendency (e.g. means) or other basic estimates (e.g. regression coefficient) AND variation (e.g. standard deviation) or associated estimates of uncertainty (e.g. confidence intervals) |
| ☐ | ☒ | For null hypothesis testing, the test statistic (e.g. $F$, $t$, $r$) with confidence intervals, effect sizes, degrees of freedom and $P$ value noted<br>*Give P values as exact values whenever suitable.* |
| ☒ | ☐ | For Bayesian analysis, information on the choice of priors and Markov chain Monte Carlo settings |
| ☒ | ☐ | For hierarchical and complex designs, identification of the appropriate level for tests and full reporting of outcomes |
| ☐ | ☒ | Estimates of effect sizes (e.g. Cohen's *d*, Pearson's *r*), indicating how they were calculated |

*Our web collection on statistics for biologists contains articles on many of the points above.*

## Software and code

Policy information about availability of computer code

| Data collection | We only used publicly available datasets and therefore did not conduct data collection. |
|---|---|
| Data analysis | For the UK Biobank GWAS analysis, we used a development branch of the Python package snipar to perform IBD state inference, parental genotype imputation and family-based GWAS (available at https://zenodo.org/records/14270274); we used the sparse LU solver for matrix inversion and the L-BFGS algorithm for variance component estimation, both implemented in the SciPy Python package[v1.7.1]; we used KING[v2.2.5] to perform IBD segment and relationship inference; we computed genetic correlations between direct genetic effect estimates from our prososed methods for height and educational attainment respectively using LDSC[v1.0.1].<br>For polygenic prediction in the Millennium Cohort Study: we used the OADP algorithm implemented in the R package bigsnpr[v1.12.2] (https://privefl.github.io/bigsnpr/) to infer sample ancestries; we used PRSCS[Jun 4, 2021] to calculate SNP weights; we used the pgs.py script in snipar to estimate direct effects of the PGIs. |

For manuscripts utilizing custom algorithms or software that are central to the research but not yet described in published literature, software must be made available to editors and reviewers. We strongly encourage code deposition in a community repository (e.g. GitHub). See the Nature Portfolio guidelines for submitting code & software for further information.

## Data

Policy information about availability of data

All manuscripts must include a data availability statement. This statement should provide the following information, where applicable:

- Accession codes, unique identifiers, or web links for publicly available datasets
- A description of any restrictions on data availability
- For clinical datasets or third party data, please ensure that the statement adheres to our policy

Summary statistics from the different estimators applied to UK Biobank data are available for download from the SSGAC data portal: https://thessgac.com/. Applications for access to the UKB data can be made on the UKB website (http://www.ukbiobank.ac.uk/register-apply/). Applications for Millennium Cohort Study data can be made by following the instructions here: https://cls.ucl.ac.uk/data-access-training/data-access/accessing-data-directly-from-cls/. 1000 Genomes phase 3 data can be downloaded using the download_1000G function provided by the bigsnpr R package.

## Research involving human participants, their data, or biological material

Policy information about studies with human participants or human data. See also policy information about sex, gender (identity/presentation), and sexual orientation and race, ethnicity and racism.

| Reporting on sex and gender | Results in this study apply to both sexes except for sex-specific phenotypes. Sex was determined based on self-reporting and genotype data as described in the documentation on the UK Biobank and MCS datasets. |
|---|---|
| Reporting on race, ethnicity, or other socially relevant groupings | For the UK Biobank data, we classified individuals into the 'white British' ancestry subsample using the flag provided by UK Biobank. For the Young et al. and unified estimator, we used individuals classified as 'White British' to make our results comparable to existing genome-wide association studies and to control for population structure.<br><br>For the the Millennium Cohort Study, we classified individuals into European (EUR) and South Asian (SAS) using the OADP and KNN algorithms. We projected individuals onto genetic principal components derived from 1000 Genomes data and classified individuals as belonging to the EUR or SAS superpopulations if all 20 of their nearest neighbors in the 1000 Genomes data were from the EUR or SAS samples. We referred to these samples as 'EUR' and 'SAS' samples. |
| Population characteristics | The UK Biobank project is a prospective cohort study with deep genetic and phenotypic data collected on approximately 500,000 individuals from across the United Kingdom, aged between 40 and 69 at recruitment. The Millennium Cohort Study is a representative sample of individuals born in the UK in the year 2000 and their parents. |
| Recruitment | Recruitment is not applicable since we used existing datasets. |
| Ethics oversight | UK Biobank has approval from the North West Multi-centre Research Ethics Committee (MREC) as a Research Tissue Bank (RTB) approval. This approval means that researchers do not require separate ethical clearance and can operate under the RTB approval (there are certain exceptions to this which are set out in the Access Procedures, such as re-contact applications). The Millennium Cohort Study has obtained ethical approval from NHS Research Ethics Committees (RECs). |

Note that full information on the approval of the study protocol must also be provided in the manuscript.

# Field-specific reporting

Please select the one below that is the best fit for your research. If you are not sure, read the appropriate sections before making your selection.

☒ Life sciences ☐ Behavioural & social sciences ☐ Ecological, evolutionary & environmental sciences

For a reference copy of the document with all sections, see nature.com/documents/nr-reporting-summary-flat.pdf

# Life sciences study design

All studies must disclose on these points even when the disclosure is negative.

| Sample size | We used genotyped individuals in the UK Biobank passing filters corresponding to specific methods:<br>For the sib-difference method, we used all genotyped individuals passing quality control filters with at least one genotyped sibling: 46,698;<br>For the robust estimator, we used all genotyped individuals passing quality control filters with at least one non-transmitted parental allele observed: 51,875;<br>For the Young et al. estimator, we used all genotyped individuals of 'White British' ancestry (identified by the UK Biobank), passing quality control filters, and with at least one genotyped first-degree relative: 44,570;<br>For the unified estimator, we used all genotyped individuals of 'White British' ancestry (identified by the UK Biobank) and passing quality control filters: 408,254.<br>For the Millennium Cohort Study, we used all genotyped individuals of either European or South Asian ancestry identified by the OADP and KNN algorithms: 10416. |
|---|---|
| Data exclusions | For analyses on UK Biobank, we filtered out individuals identified by UK Biobank as having excess relatives, excess heterozygosity, sex chromosome aneuploidy, and excess genotype missingness. |

| | |
|---|---|
| | For the Millennium Cohort Study, we restricted our sample to individuals of European or South Asian ancestry identified by the OADP and KNN algorithms, to make our analyses comparable to existing studies on cross-ancestry polygenic prediction. |
| Replication | Our UK Biobank analysis was performed to show the relative power of the different estimators in a large-scale biobank. Direct replication of these results is not relevant to the conclusions of our study. However, our Millennium Cohort Study analysis demonstrates that our UK Biobank analysis produced results with external validity by demonstrated statistically significant out-of-sample prediction ability for genetic predictors derived from our UK Biobank analysis. |
| Randomization | Genetic materials are randomized during meioses; this randomization is used in family-based GWAS designs to remove confounding. |
| Blinding | Blinding is not applicable since we did not compare experimental groups. |

# Reporting for specific materials, systems and methods

We require information from authors about some types of materials, experimental systems and methods used in many studies. Here, indicate whether each material, system or method listed is relevant to your study. If you are not sure if a list item applies to your research, read the appropriate section before selecting a response.

## Materials & experimental systems

| n/a | Involved in the study |
|---|---|
| ☒ | Antibodies |
| ☒ | Eukaryotic cell lines |
| ☒ | Palaeontology and archaeology |
| ☒ | Animals and other organisms |
| ☒ | Clinical data |
| ☒ | Dual use research of concern |
| ☒ | Plants |

## Methods

| n/a | Involved in the study |
|---|---|
| ☒ | ChIP-seq |
| ☒ | Flow cytometry |
| ☒ | MRI-based neuroimaging |

