## [Peer Review File · Nature Genetics]

Family-based genome-wide association study designs for increased power and robustness

Corresponding Author: Mr Junming Guan

Version 0:

Decision Letter:

18th Jul 2023

Dear Junming,

Your Technical Report, "Novel estimators for family-based genome-wide association studies increase power and robustness" has now been seen by 2 referees. You will see from their comments below that while they find your work of interest, some important points are raised. We are interested in the possibility of publishing your study in Nature Genetics, but would like to consider your response to these concerns in the form of a revised manuscript before we make a final decision on publication.

In brief, the two reviewers are positive and we think there is a path to publication, but they do have some comments that will require further analyses.

Reviewer #1 says their comments are to further improve the quality of your study. Their most important request is for expanded simulations that would further clarify the breadth of utility of the proposed estimators.

Referee #2 is not so explicitly positive, but also seems to appreciate the aims of your study. Like Reviewer #1, their major request is to help clarify the utility of the estimators, but they ask for expanded analysis of the empirical data to illustrate such.

In our reading these requests all seem reasonable and achievable without requiring a substantial expansion of the work in its current form, so we hope you will be able to fulfill them all.

To guide the scope of the revisions, the editors discuss the referee reports in detail within the team, including with the chief editor, with a view to identifying key priorities that should be addressed in revision and sometimes overruling referee requests that are deemed beyond the scope of the current study. We hope that you will find the prioritized set of referee points to be useful when revising your study. Please do not hesitate to get in touch if you would like to discuss these issues further.

We therefore invite you to revise your manuscript taking into account all reviewer and editor comments. Please highlight all changes in the manuscript text file. At this stage we will need you to upload a copy of the manuscript in MS Word .docx or similar editable format.

*2) If you have not done so already please begin to revise your manuscript so that it conforms to our Technical Report format instructions, available

[here](http://www.nature.com/ng/authors/article_types/index.html).

*3) Include a revised version of any required Reporting Summary: <https://www.nature.com/documents/nr-reporting-summary.pdf>

Link Redacted

Sincerely,

Michael Fletcher, PhD
Senior Editor, Nature Genetics

ORCID: 0000-0003-1589-7087

Referee expertise: both work in statistical genetics and have expertise in non-standard GWAS designs.

Reviewers' Comments:

Reviewer #1:

Remarks to the Author:

In this study Guan et al. extend previous methods developed by Young et al. (2022) – hereafter referred to as Y22 – to combine data from family and unrelated individuals and increase robustness of family-based imputation to biases induced by population stratification. Overall, the manuscript is well written and I only have comments to clarify aspects of the paper and thereby help to improve its quality.

INTRODUCTION

"causal effects of alleles (that are physically linked with the focal allele)" – I'm not sure what is meant here. Are the authors trying to say that GWAS effects are not causal effects but reflect LD between tested and causal alleles?

"but studies have shown that this often fails to remove all confounding" – Can the authors clarify confounding of what with what? Confounding implies that GWAS effects may not be the main quantity of interest. It would be useful to clarify this here.

POPULATION STRUCTURE ROBUST ESTIMATOR

Fst is usually defined between (or across) populations. Can the authors clarify how is Fst defined here? Did they use the Hill & Weir definition relative to an ancestral population? I understand that they simulated/considered sub-groups within their analysed sample but this may not be clear to all readers.

The authors simulate allele frequencies under the Balding-Nichols model using a Beta distribution with parameters equal to $(1-F_{st})/(2F_{st})$. Is this implying that allele frequencies in the ancestral population are all equal to 0.5?

I think that simulations always serve a purpose, which here was to illustrate theoretical results. However, the readers needs to be reassured that the scenarios considered do not hide other interesting behaviours that might occur with real data. Therefore, can the authors show/prove that their conclusions are robust to the choice of allele frequencies distributions? Are some of the biases observed with the unified estimator bigger when allele frequencies are smaller (e.g., ~0.1)?

Also, the simulations assume that 50% of the phenotypic variance is explained by stratification? This seems quite extreme. I

understand that extreme is good to show that your methods are robust but this particular scenario would also be quite favourable to a simple PCA analysis. If $F_{st}=0.1$ and 50% of trait variance is due to stratification then I'd expect standard GWAS fitting PCs to perform quite well. On a related note, I could not figure out whether the standard GWAS was a simple linear regression or if it included PCs or even was fitting a LMM?

I guess my questions are (i) how well do these estimators perform relative to standard GWAS with the "best" existing corrections? (ii) What happens under less extreme scenarios?

Why was robust estimator referred to with quotation marks?

COMPARISON OF ESTIMATORS

Can the authors justify why they quantified biases in terms of non-sampling variance? I thought that the statistical literature had more straightforward ways to quantify biases.

FIGURE 5: I was wondering if the behaviour reported in that figure was dependent on data dimension (i.e., number of individuals and SNPs). In particular, I'm curious to know if the bias observed for the unified estimator reflects some phase transition in the distribution of eigenvalues of the genotype matrix. This sort of behaviour is not uncommon and reflects inherent properties of random matrices. I'm not sure what would be the right experiment to conduct here but can the authors comment on that? Do they think that other situations could lead to large biases if even F_{st} is smaller?

APPLICATION TO REAL DATA

Can the authors directly quantify sample overlap (or relatedness) between UKB and MCS? What is the ancestry composition of MCS?

If I understand well Figure 7 then the PGI based on population effects has a larger accuracy in SAS than in EUR (at least for height). This seems to contradict what was previously reported in the literature. What could be the explanation for? Would the authors get the same effects if adjusting analyses in the MCS using principal components?

Another quantity of interest for Figure 7 would be the regression slopes for each PGI. This is important to measure shrinkage of effect sizes and would give an indication about biases in real data.

DISCUSSION

How would the robust estimator perform in an admixed population?

Other comments

I could not see Table 3 in the main text. So I think there is a typo. Table 3 should be Table 2, right?

The reader needs a bit more information about the phenotypes (at least in the UK Biobank). What field corresponds to the phenotypes used here. For example, I don't think there is a UK Biobank field named "cognitive ability". Can the authors specify which ones were used and whether only data from baseline were used if combined across visits.

Can the authors report/mention the computational time of their methods somewhere in a table?

Reviewer #2:

Remarks to the Author:

Standard (population/non-family) GWAS provide the highest power for detecting genetic associations but effect estimates are impacted by assortative mating, population stratification and indirect genetic effects. Within-family GWAS provide better estimates of direct genetic effects but have much lower statistical power.

Guan et al provide an interesting technical report on improving the power of within-family GWAS by parental genotype imputation, expanding on previous work by the authors.

In previously published work the authors presented an approach imputing parental genotypes for individuals with first degree relatives in the study.

In this work the approach has been extended to include two additional models:

robust model - excluding IBD2 siblings where imputation was more challenging in previous work.

unified model - including singletons (individuals without close relatives in the study).

The authors use simulations and theory to illustrate how improved statistical power across models comes with the risk of additional population stratification bias with the exception of the robust model. They then perform empirical analyses in UK Biobank (GWAS) and in the MCS (polygenic prediction).

The manuscript is generally well-written and although I haven't used the software package (and updates are coming), the software (via GitHub) looks polished and user-friendly. Please find my detailed comments below.

1)

I'm convinced that the robust model is preferable to the sibling differences model with higher statistical power (10-25%) at the cost of computational time for the imputation.

However, I'm not so convinced about the practical usefulness of the unified model because of potential susceptibility to population stratification.

Generally researchers interested in within-family analyses are analysing phenotypes highly susceptible to non-direct genetic effects (height, educational attainment).

Therefore, it seems to me that the unified estimator represents a compromise which has both advantages and disadvantages over the sib differences/robust and standard GWAS models.

If going for statistical power, a researcher might prefer to use the standard GWAS and if going for less bias might prefer sib differences/robust estimators.

To help understand the strengths of the unified estimator and to provide a more measured summary:

a) Could the authors provide further empirical investigation of how the UKB results from the different models are affected by "non-direct" genetic effects?

Maybe treating the sib difference/robust estimates as "the truth" and then comparing the GWAS effect sizes between the different models for height/educational attainment.

Could look at top hits and random variants.

In theory, the unified effect estimates should be on average the same but with lower standard errors.

This would provide an empirical estimate of how close the unified estimator results are to the robust estimates in real data.

If the "bias" is near-zero that would support the use of the unified estimator.

b) Abstract

The abstract mentions "increasing effective sample size for direct genetic effects by 44.5%-106.7% compared to using genetic differences between siblings" which I assume is from the unified model.

The results mentions a more modest "effective sample size of between 10.8% (height) and 27.4%" presumably from the robust model.

Given the unified model is potentially susceptible to population stratification please clarify that the increase comes with risk of bias or present the robust model increases.

Technically the standard GWAS model would increase effective sample size for direct genetic effects substantially (at a cost)!

2) Genome-wide versus autosomal.

Could the authors clarify if their framework is only applicable to the autosomes?

In particular, the X chromosome contains ~850 protein-coding genes so is potentially important, but is often omitted from "GWAS".

Perhaps out of scope but if straightforward (and necessary) would it be possible to extend the model/software for the X chromosome?

If not please acknowledge as a limitation.

Minor comments:

1) Introduction: some minor grammatical tweaks

"Adjustment for genetic principal components and linear mixed models (LMMs) reduce" (reduces)

"Family-based GWAS uses" (use)

2) The 0.05 threshold for IBD sparsity.

Could the authors please elaborate on how sensitive the results are to this threshold.

How would results change with a lower threshold?

3) Imputed versus sequence data.

UK Biobank and other studies are performing WGS on study participants.

I suspect WGS data wouldn't have much of an advantage over imputed data for this framework but interested for the author's thoughts.

4) Please ensure the GitHub README is updated to provide guidance on the different models that can be run to ensure a user-friendly experience.

Reviewer #3:

None

Version 1:

Decision Letter:

2nd Oct 2024

Dear Dr. Guan,

Your Technical Report, "Novel estimators for family-based genome-wide association studies increase power and robustness" has now been seen by 2 referees. You will see from their comments below that while they find your work of interest, there are still a few comments raised. We remain interested in the possibility of publishing your study in Nature Genetics, but would like to consider your response to these concerns in the form of a revised manuscript before we make a final decision on publication.

Briefly, Reviewer #1 is satisfied and has no further comments, but Referee #2 has a few. We think these are minor and may not require further review, but we noted the question on how your methods operate in the ultra-low MAF range. Given the increasing interest in studying these (ultra-)rare variants given the growing scale of biobanks, we think it would be useful to have clarification on this point before a final decision.

To guide the scope of the revisions, the editors discuss the referee reports in detail within the team, including with the chief editor, with a view to identifying key priorities that should be addressed in revision and sometimes overruling referee requests that are deemed beyond the scope of the current study. We hope that you will find the prioritized set of referee points to be useful when revising your study. Please do not hesitate to get in touch if you would like to discuss these issues further.

We therefore invite you to revise your manuscript taking into account all reviewer and editor comments. Please highlight all changes in the manuscript text file. At this stage we will need you to upload a copy of the manuscript in MS Word .docx or similar editable format.

*2) If you have not done so already please begin to revise your manuscript so that it conforms to our Technical Report format instructions, available

[here](http://www.nature.com/ng/authors/article_types/index.html).

*3) Include a revised version of any required Reporting Summary: <https://www.nature.com/documents/nr-reporting-summary.pdf>

Link Redacted

We hope to receive your revised manuscript within four to eight weeks. If you cannot send it within this time, please let us know.

Nature Genetics is committed to improving transparency in authorship. As part of our efforts in this direction, we are now requesting that all authors identified as 'corresponding author' on published papers create and link their Open Researcher and Contributor Identifier (ORCID) with their account on the Manuscript Tracking System (MTS), prior to acceptance. ORCID helps the scientific community achieve unambiguous attribution of all scholarly contributions. You can create and link your

ORCID from the home page of the MTS by clicking on 'Modify my Springer Nature account'. For more information please visit www.springernature.com/orcid.

Sincerely,

Michael Fletcher, PhD
Senior Editor, Nature Genetics
ORCID: 0000-0003-1589-7087

Reviewers' Comments:

Reviewer #1 (Remarks to the Author):

I thank the authors for thoroughly addressing all my comments. The additional work on the effect of admixture was particularly insightful. I have no additional comments or suggestions.

Reviewer #2 (Remarks to the Author):

The authors have well-addressed my previous comments. The new updates to the manuscript such as the admixture-robust estimator look great.

I have only a few minor comments.

1. In the introduction you discuss different contributors to genetic association estimates. Could also mention selection bias relating to non-randomness of the dataset. Participation bias in UKB is one element which people have investigated. There's also case-only analyses (e.g. disease progression GWAS) where estimates can be distorted after stratifying on a heritable factor. Higher disease polygenic scores associated with lower mortality in cases is a classic collider example.
2. Introduction: 'overestimation of heritability and the traits' shared genetic architectures.' Could it also be underestimation under some theoretical model? Maybe mis-estimation or biased estimates is safer.
3. Spurious inferences of disease causes by mendelian randomisation'. Suggest rephrasing to 'spurious inferences in Mendelian randomisation analyses'. I can't think of examples where population based MR has been shown to be biased by AM/pop strat/ IGE for a disease outcome. I think most examples were with 'social/behavioural' outcome phenotypes. For example, the reference is BMI/Height on Educational Attainment. Population based GWAS of complex disease benchmark very well for successful drug targets and Mendelian disease genes.
4. Rare variants: I'm trying to wrap my head around how different within-family models work for rare variants (MAF < 0.01). Figure 2B has the theoretical gain in effective N across the MAF frequency between different models. This is the relative power, assume the absolute power/precision of within-family GWAS models plummet with allele frequency as we require heterozygosity within-families? e.g. a variant has MAF of 0.0001 in N = 50K. This would have MAC of 10 and only a subset of relateds would have differences. Rare variants more likely to co-occur within families lowering heterozygosity so power may be even lower than expected? No major action required, interested for thoughts on how the models perform in the rare/ultra-rare space.

Version 2:

Decision Letter:

Our ref: NG-TR62624R1

24th Oct 2024

Dear Junming,

Thank you for submitting your revised manuscript "Novel estimators for family-based genome-wide association studies increase power and robustness" (NG-TR62624R1).

We've made an editorial check of the changes and we are satisfied that they do not require further review, such that we'll be happy in principle to publish it in Nature Genetics, pending minor revisions to satisfy any final requests and to comply with our editorial and formatting guidelines.

If the current version of your manuscript is in a PDF format, please email us a copy of the file in an editable format (Microsoft

Word or LaTeX)-- we can not proceed with PDFs at this stage.

Sincerely,

Michael Fletcher, PhD
Senior Editor, Nature Genetics
ORCID: 0000-0003-1589-7087

Notification of major changes for all reviewers

First, we want to thank the reviewers for their constructive and perspicacious comments. We believe we have been able to successfully address all criticisms raised by the reviewers while substantially strengthening the manuscript. We have made some major changes to our manuscript that we outline here for the convenience of all reviewers. In addition to the major changes we outline here, we give a point-by-point response to reviewer comments below.

New robust estimator

Reviewer #1 made a comment about how the robust estimator would perform in admixed samples. This led us to investigate this both theoretically and in simulations (Supplementary Note Section 2.3.3). We found that the robust estimator was biased when paternal and maternal allele frequencies are different, as is the case in recent admixture between parents of distinct genetic ancestries. In response to this finding, we decided to develop another estimator that is robust to both population stratification and admixture. We developed such an estimator by performing uniparental regressions in the samples where we have observed only one non-transmitted allele and we know the parent of origin. This new estimator, which we also call the ‘robust estimator’, has slightly decreased power compared to the old estimator, which we now term the ‘non-transmitted estimator’, but is robust to admixture, whereas the ‘non-transmitted estimator’ is robust to an island model of population structure, but not recent admixture. We give here the relevant section of the manuscript pertaining to the new typology of estimators:

Population-structure-robust estimator

The imputation proposed by Young et al.³ uses the overall allele frequency in the sample to impute unobserved parental alleles. The imputation becomes biased when there is population structure as the imputation does not account for variation in allele frequencies across subpopulations³. Young et al. showed that, in an island model of population structure, the estimator of DGEs from sibling pairs with parental genotypes imputed from phased data tends to $\hat{\delta} = \delta + c\alpha$, where c is a function of Wright's F_{st} (the proportion of variation at a locus due to between-population differences in allele frequencies). When F_{st} is small, $c \approx F_{st}/2$, implying the bias, $c\alpha$, will be negligible for European genetic ancestry samples, where F_{st} has been estimated to be on the order of 10^{-3} (ref²²). In contrast, under the island model of population structure, standard GWAS estimates $\beta = \delta + \frac{1+3F_{st}}{1+F_{st}}\alpha > \delta + \alpha$.

Young et al.³ proposed an alternative, imputation-based estimator for sibling-pair data that they argued should not be biased by population structure. This estimator partitioned the sibling pairs based on their IBD state and performed separate regressions for sibling pairs in IBD states 0 and 1 followed by an inverse-variance-weighted meta-analysis of the DGE estimates. Young et al.³ showed this is more powerful than the sibling-difference estimator, having a relative effective sample size $1 + \frac{1-r}{6(1+r)}$ times greater, where r is the correlation of siblings' residuals. However, this estimator has a smaller effective sample

size than the primary estimator considered in Young et al.³, which includes sibling pairs in IBD2, at the cost of bias due to population structure.

Group	Example genotype data types	Non-transmitted (NT) alleles observed	Non-transmitted estimator regression	Robust estimator regression
Maternal NT	Mother-child pairs Mother and sibling pair in IBD2	maternal	$y_{ij} \sim g_{ij} + \hat{g}_{p(i)} + g_{m(i)}$	$y_{ij} \sim g_{ij}^m + g_{m(i)}$
Paternal NT	Father-child pairs Father and sibling pair in IBD2	paternal	$y_{ij} \sim g_{ij} + g_{p(i)} + \hat{g}_{m(i)}$	$y_{ij} \sim g_{ij}^p + g_{p(i)}$
Both NT	Sibling pairs in IBD0 Parent-offspring trios	paternal and maternal	$y_{ij} \sim g_{ij} + g_{par(i)}$	$y_{ij} \sim g_{ij} + g_{par(i)}$
One NT	sibling pairs in IBD1 without genotyped parents	paternal or maternal	$y_{ij} \sim g_{ij} + \hat{g}_{par(i)}$	$y_{ij} \sim g_{ij} + \bar{g}_{sib(i)}$

Table 1. Groups and regressions for the non-transmitted robust estimator. We partition the sample with at least one non-transmitted (NT) parental allele observed into four groups (Supplementary Figure 2). By performing the regressions separately in each group, we obtain consistent estimates of direct genetic effects from each group, even when there is population structure and/or admixture (Supplementary Note Sections 2.3 & 2.4). For the regression column, y_{ij} is the phenotype of sibling j in family i ; g_{ij} the corresponding genotype; g_{ij}^m and g_{ij}^p are the maternally and paternally transmitted alleles; $g_{p(i)}$ and $g_{m(i)}$ are the paternal and maternal genotypes; $g_{par(i)} = g_{p(i)} + g_{m(i)}$; a caret indicates a genotype that has been imputed from phased data as in Young et al.³; e.g., $\hat{g}_{par(i)}$ refers to the imputed sum of parental genotypes. $\bar{g}_{sib(i)}$ is the mean genotype among all siblings in family i .

Here, we developed a generalization of the robust estimator proposed by Young et al. so that it can handle all possible data types, rather than just sibling pairs: it partitions the sample not on IBD state but on which parental alleles that were not transmitted to the focal, phenotyped individual (proband) we have observed: this gives four groups (Table 1 and Supplementary Figure 2) depending on whether we have observed one or both non-transmitted parental alleles, and if only one has been observed, whether the non-transmitted allele is from the mother, father, or unknown (as for sibling pairs in IBD1 without genotyped parents). We call this estimator the ‘non-transmitted estimator’. We prove theoretically that this estimator gives consistent estimators of DGEs under an island model of population structure (Supplementary Note Section 2.3.2).

However, we found that the non-transmitted estimator can give biased estimates when there are differences in allele frequencies between mothers and fathers (Supplementary Note Section 2.3.3), as in admixed samples. We therefore sought to develop an estimator that avoids all bias due to population structure or admixture while maximizing power: we call this the ‘robust estimator’, as it uses only the random variation in offspring genotype given parental genotype, which is the principle underlying the properties of FGWAS with fully observed parental genotypes.

This estimator — like the non-transmitted estimator — partitions the sample based on which non-transmitted alleles have been observed (Table 1). However, it differs in the regressions it performs. It performs uniparental regressions for the samples with one parental non-transmitted allele observed when the parent-of-origin is known; for example, for a sample with a mother genotyped, the regression is performed on the maternally transmitted allele and the mother’s genotype, thereby only using the random

variation in maternally inherited allele given maternal genotype to estimate the DGE. When both parent and offspring are heterozygous, phased data is required to determine the parent-of-origin³. For sibling pairs in IBD1 without genotyped parents, the parent-of-origin of alleles is unknown, and we perform a regression controlling for the mean genotype in the sibship, equivalent to sib-difference regressions (Supplementary Note Section 2.5). However, for sibling pairs in IBD1 with a genotyped parent, we partition the sample based on whether the allele shared by the siblings is from the observed parent or the missing parent: when the shared allele is from the missing parent, we perform uniparental regressions using the observed parental genotype and the alleles inherited by the siblings from that parent (e.g. a sibling pair in IBD1 with a genotyped mother would be placed in the maternal NT group when the shared allele is from the father); but when the shared allele is from the observed parent, we are able to fully recover the missing parent's alleles and place the siblings in the both NT group. (See Supplementary Note Section 5.2 from Young et al.³ for further details on determining shared alleles for cases with one parent and multiple offspring with observed genotypes.)

The advantage of the robust estimator over using genetic differences between siblings and/or samples with both parents genotyped observed is that it enables optimal use of samples with a single parent genotyped while not using any allele frequency information that can introduce bias in structured populations. When only siblings are genotyped with no parents, it becomes equivalent to using genetic differences between siblings (see Methods and Supplementary Note Section 2.5.1); and when all samples have both parents genotyped, it reduces to fitting model (2) with fully observed parental genotypes.

More realistic population structure simulations

In response to Reviewer #1's criticism of our simulations' lack of realism, we have performed simulations of more complex population structure scenarios that also give some insight into how population stratification confounding works in a Balding-Nichols model of population structure. These simulations do not alter our previous conclusions; in fact, they strengthen them, showing the general robustness of the family-based GWAS (FGWAS) estimators to complex population structure. We give here the description of the novel simulations:

To investigate a more complex and realistic scenario, we performed additional simulations with 100 subpopulations (instead of two, as above) and with ancestral minor allele frequencies, f , drawn from a distribution with density proportional to $1/f$ for $0.05 < f < 0.5$. We chose this distribution as it reflects the distribution of allele frequencies for a randomly mating population with constant effective size²⁵. As above, we simulated no true DGEs, with 50% of the phenotypic variance explained by subpopulation membership, so any statistical deviation from the null is evidence of population stratification. We display results for the estimators in Figure 4, where we have also included standard GWAS with adjustment for 20, 50, and 99 inferred genetic principal components (PCs) (Methods). (Since there are 100 subpopulations, 99 PCs should be sufficient to separate all subpopulations if inferred correctly^{7,26}.) We did not find statistically significant evidence (here and below, P-value < 0.05) for population stratification bias for any of the FGWAS estimators (sib-difference/robust, non-transmitted, Young et al., and unified). The reason we did not find statistically significant

evidence for bias when simulating 100 subpopulations (unlike with two) is likely because the expected magnitude of the population stratification confounding goes down with the number of subpopulations (Methods), so that with 100 subpopulations the bias becomes statistically undetectable from 20,000 SNPs.

Figure 5. Bias-variance tradeoff on simulated sibling pairs and singletons. The simulated datasets used in Figure 3 are used for this demonstration: 2,000 independent sibling pairs and 18,000 singletons in each of two subpopulations differentiated different levels of F_{st} (a)-(d) (Methods). Effective sample size (x-axis) is defined relative to that of the sib-difference estimator (Table 2). Bias (y-axis) is measured as the non-sampling variance (Methods) relative to that for standard GWAS with $F_{st} = 0.001$. See Supplementary Figures 3 & 4 for plots including the standard GWAS estimator and a sibling-only scenario (i.e., no singletons).

However, we found statistically significant evidence of bias for standard GWAS when $F_{st} > 0$ regardless of how many PCs we controlled for, with one exception: when $F_{st} = 0.1$ and we controlled for 99 PCs. The explanation for these results is likely because it is difficult to infer 99 PCs correctly without very large sample sizes when population structure is subtle ($F_{st} \leq 0.01$) but becomes easier when population structure is stronger (as in the $F_{st} = 0.1$ scenario). This is related to the known phase transition whereby it becomes possible to accurately infer latent factors (i.e., subpopulation membership) that structure random matrices (e.g. SNP genotype matrices) once the sample size passes a certain threshold, depending on the strength of those latent factors⁷. In real-world genetic data, population structure likely exists on multiple scales (reflecting both recent and ancient structure) with genetic principal components only imperfectly capturing subtle and recent structure^{2,10}.

We also give here the new Methods section that gives details on the more complex simulations and how population stratification confounding works in a Balding-Nichols model:

Simulations of structured populations

For different levels of F_{st} , we generated K subpopulations. We simulated 20,000 SNPs from binomial distributions, where subpopulation allele frequencies were drawn from the Balding-Nichols model²³: $f_k \sim \text{Beta}\left(\frac{1-F_{st}}{F_{st}} f, \frac{1-F_{st}}{F_{st}} (1-f)\right)$, where f_k is the allele frequency in subpopulation k and $f = E_k[f_k]$ is the overall allele frequency.

We investigated two different scenarios: a simple scenario with two subpopulations ($K = 2$), where each subpopulation has 20,000 families each with two siblings, and all overall allele frequencies are 0.5 ($f = 0.5$); and a more complex scenario with 100 subpopulations ($K = 100$), each with 1,000 families, and overall allele frequencies drawn from a distribution proportional to $1/f$ for $0.5 > f > 0.05$.

We generated phenotypes with 50% of the phenotypic variance attributed to subpopulation phenotype means μ_k that we sampled independently from a mean-zero normal distribution: $\mu_k \sim N(0, \sigma_\mu^2)$. The remaining 50% of the phenotypic variance was attributed to random Gaussian noise, implying a correlation between siblings' phenotypes of 0.5. There are no causal effects (including DGEs) of the genotypes in this simulation, so any deviation from the null distribution is evidence of bias due to population stratification.

The average confounding bias due to population stratification in this model is zero, but the average magnitude of the confounding is non-zero for a finite number of subpopulations K . To see this, consider the overall covariance between the genotype g at a SNP l and phenotype Y , which is the covariance between the subpopulation genotype means ($2f_k$) and subpopulation phenotypic means (μ_k):

$$\text{Cov}(g, Y) \approx \sum_k p_k 2(f_k - \bar{f}) \mu_k,$$

where f_k is the allele frequency in subpopulation k , $\bar{f} = \sum_k p_k f_k$ is the overall allele frequency, and p_k is the fraction of families in subpopulation k . The regression coefficient of genotype onto phenotype is therefore

$$\beta_f = \frac{\text{Cov}(g, Y)}{\text{Var}(g)} \approx \frac{\sum_k p_k 2(f_k - f) \mu_k}{\text{Var}(g)}$$

Since the allele frequencies and subpopulation phenotype means are sampled independently, the regression of phenotype onto genotype has expectation zero across the SNPs but varies around zero due to the finite number of subpopulations. We quantify this through the expected squared regression coefficient across SNPs with overall allele frequency f :

$$E[\beta_f^2] \approx \frac{\sum_k 4p_k^2 F_{st} f(1-f)\sigma_\mu^2}{\text{Var}(g)^2},$$

where we have used the variance of the allele frequencies from the Balding-Nichols model ($F_{st}f(1-f)$) to obtain this. If we assume equal subpopulation sizes, then $p_k = 1/K$ and

$$E[\beta_f^2] \approx \frac{4F_{st}f(1-f)\sigma_\mu^2}{K\text{Var}(g)^2} = \frac{F_{st}\sigma_\mu^2}{Kf(1-f)(1+F_{st})^2},$$

where we have used the fact³ that $\text{Var}(g) = 2f(1-f)[1+F_{st}]$. The expected magnitude of the bias therefore decreases (towards zero) with the number of subpopulations, holding f , F_{st} , and σ_μ^2 constant. If we consider the expected phenotypic variance explained (in a regression sense, i.e. the R^2) by each variant, it does not depend upon overall allele frequency:

$$\frac{\text{Var}(g)E[\beta_f^2]}{\text{Var}(Y)} \approx \frac{2F_{st}\sigma_\mu^2}{K(1+F_{st})\text{Var}(Y)}.$$

For the simulations involving the unified estimator, we sought to mimic the fact that large biobank datasets such as the UK Biobank consist mostly of singletons. For 90% of families, we randomly removed one sibling to obtain singletons, leaving 10% of families with two siblings. The sibling-difference/robust, non-transmitted, and Young et al. estimators were applied to the 10% of families with intact sibling pairs, while the unified estimator and standard univariate GWAS were applied to the combined sample of singletons and sibling pairs.

We also examine the performance of the estimators in a sibling-pair-only scenario: i.e., 20,000 genotyped and phenotyped sibling pairs in each subpopulation (Supplementary Figure 4). We applied the estimators to the resulting 40,000 sibling pairs. Note that in this scenario, there are no singletons, and the unified estimator is equivalent to the Young et al. estimator.

Reviewers' Comments:

Reviewer #1:

Remarks to the Author:

In this study Guan et al. extend previous methods developed by Young et al. (2022) – hereafter referred to as Y22 – to combine data from family and unrelated individuals and increase robustness of family-based imputation to biases induced by population stratification. Overall, the manuscript is well written and I only have comments to clarify aspects of the paper and thereby help to improve its quality.

We are glad the reviewer found the manuscript to be well written, and we believe we have been able to improve the clarity of presentation thanks to the reviewers' comments.

INTRODUCTION

“causal effects of alleles (that are physically linked with the focal allele)” – I’m not sure what is meant here. Are the authors trying to say that GWAS effects are not causal effects but reflect LD between tested and causal alleles?

Yes, we meant that GWAS can pick up causal effects of tested alleles and those alleles in LD with the tested alleles. Thank you for pointing out the confusion that our phrasing could have caused. To make this point clearer, we have now modified the sentence to read “causal effects of alleles — both of the tested variant and those in linkage disequilibrium (LD) with the tested variant — carried by the individual on the individual (direct genetic effects)”.

“but studies have shown that this often fails to remove all confounding” – Can the authors clarify confounding of what with what? Confounding implies that GWAS effects may not be the main quantity of interest. It would be useful to clarify this here.

The reviewer is correct to highlight that confounding only makes sense as a concept with respect to some quantity of interest. Although sometimes researchers have framed the goal of GWAS as simply to find replicable genotype-phenotype associations, or to create powerful genetic predictors, many of the downstream applications of GWAS data implicitly assume that GWAS summary statistics are something close to confounding-free estimates of causal effects of alleles in an individual on that individual. We have reworked the paragraph to clarify what we mean by confounding and what the consequences of such confounding may be for downstream applications:

If we consider the goal of GWAS to be estimation of direct genetic effects (DGEs), then the other contributors to GWAS estimates — IGEs and other factors (both genetic and environmental) with which the tested variant is associated with due to non-random mating — can be considered as confounds. Adjustment for genetic principal components and linear mixed models (LMMs) reduces confounding due to population stratification^{7,8} and AM¹¹, but studies have shown residual confounding often remains^{3,8,9,11}. The consequences of this bias include: 1) overestimation of heritability and the traits’ shared genetic architectures (through genetic correlation estimates)^{12–15}; 2) spurious inferences of disease causes by Mendelian Randomization¹⁶; 3) bias in polygenic indexes (PGIs, also called polygenic scores) that may contribute to the drop in predictive accuracy when predicting across genetic ancestries¹⁷; and 4) spurious inferences of natural selection^{9,15,18}.

POPULATION STRUCTURE ROBUST ESTIMATOR

Fst is usually defined between (or across) populations. Can the authors clarify how is Fst defined here? Did they use the Hill & Weir definition relative to an ancestral population? I understand that they simulated/considered sub-groups within their analysed sample but this may not be clear to all readers.

Thank you for pointing out that our definition of Fst was not clear to readers. We use Wright’s Fst throughout our manuscript, where this is defined as the proportion of genotype variation due

to between population allele frequency differences. We have now added this to the first paragraph of the Population Structure Robust Estimator subsection where we first introduce Wright's F_{st} : "...Wright's F_{st} (the proportion of variation at a locus due to between population differences in allele frequencies)."

The authors simulate allele frequencies under the Balding-Nichols model using a Beta distribution with parameters equal to $(1-F_{st})/(2F_{st})$. Is this implying that allele frequencies in the ancestral population are all equal to 0.5?

Yes, the reviewer is correct that the simulations in the initial submission were all based on ancestral allele frequency equal to 0.5. See below for a more complete response.

I think that simulations always serve a purpose, which here was to illustrate theoretical results. However, the readers needs to be reassured that the scenarios considered do not hide other interesting behaviours that might occur with real data. Therefore, can the authors show/prove that their conclusions are robust to the choice of allele frequencies distributions? Are some of the biases observed with the unified estimator bigger when allele frequencies are smaller (e.g., ~ 0.1)?

Also, the simulations assume that 50% of the phenotypic variance is explained by stratification? This seems quite extreme. I understand that extreme is good to show that your methods are robust but this particular scenario would also be quite favourable to a simple PCA analysis. If $F_{st}=0.1$ and 50% of trait variance is due to stratification then I'd expect standard GWAS fitting PCs to perform quite well. On a related note, I could not figure out whether the standard GWAS was a simple linear regression of if it included PCs or even was fitting a LMM?

I guess my questions are (i) how well does these estimators perform relative to standard GWAS with the "best" existing corrections? (ii) What happens under less extreme scenarios?

We thank the reviewer for raising the above points about our simulations. We agree that the simulations were not realistic: they were intended to demonstrate the behaviour of the estimators in certain simple scenarios that are easy to understand. However, as the reviewer points out, this may leave the reader in doubt about whether our conclusions hold in more realistic scenarios. In order to answer the points raised by the reviewer, we have performed a new set of simulations that: 1) simulate more complex structure due to 100 subpopulations (compared to the two subpopulation simulations we performed earlier); 2) uses a more realistic distribution of ancestral allele frequencies, f , which sampled from a distribution with density proportional to $1/f$ (as in a random-mating population of constant effective size) for $f > 0.05$; and 3) examines the performance of standard GWAS when augmented with different numbers of principal components to control for population stratification.

We include the new text and figure here for your convenience:

To investigate a more complex and realistic scenario, we performed additional simulations with 100 subpopulations (instead of two, as above) and with ancestral minor allele frequencies, f , drawn from a distribution with density proportional to $1/f$ for

$0.05 < f < 0.5$. We chose this distribution as it reflects the distribution of allele frequencies for a randomly mating population with constant effective size²⁵. As above, we simulated no true DGEs, with 50% of the phenotypic variance explained by subpopulation membership, so any statistical deviation from the null is evidence of population stratification. We display results for the estimators in Figure 4, where we have also included standard GWAS with adjustment for 20, 50, and 99 inferred genetic principal components (PCs) (Methods). (Since there are 100 subpopulations, 99 PCs should be sufficient to separate all subpopulations if inferred correctly^{7,26}.) We did not find statistically significant evidence (here and below, P-value < 0.05) for population stratification bias for any of the FGWAS estimators (sib-difference/robust, non-transmitted, Young et al., and unified). The reason we did not find statistically significant evidence for bias when simulating 100 subpopulations (unlike with two) is likely because the expected magnitude of the population stratification confounding goes down with the number of subpopulations (Methods), so that with 100 subpopulations the bias becomes statistically undetectable from 20,000 SNPs.

Figure 4. Bias and non-sampling variance of estimators under complex population structure. We simulated four different population with different levels of structure, as measured by Wright’s F_{st} . Each population consisted of 100 equally sized subpopulations with 100 independent sibling pairs and 900 singletons. Allele frequencies for the subpopulations were simulated from the Balding-Nichols²³ model with ancestral allele frequencies, f , drawn from a distribution proportional to $1/f$ (Methods). We simulated phenotypes with no causal genetic effects but where subpopulation membership explained 50% of the phenotypic variance, so that any deviation from the null distribution indicates population stratification confounding. For standard GWAS estimators, we inferred principal components and performed standard GWAS adjusting for different numbers of principal components (Methods): 0, 20, 50, and 99. (a) Mean of squared Z-statistics across 20,000 SNPs for the four estimators, which are expected to be above 1 (dashed-line) when there is bias due to population stratification; (b) non-sampling variances (see Equation 2 in Methods) of the estimators relative to the that observed for standard GWAS with $F_{st} = 0.001$, which gives a measure of the magnitude of bias due to population stratification, with values above 0 indicating bias. Error bars display 95% confidence intervals over 20,000 SNPs.

However, we found statistically significant evidence of bias for standard GWAS when $F_{st} > 0$ regardless of how many PCs we controlled for, with one exception: when $F_{st} = 0.1$ and we controlled for 99 PCs. The explanation for these results is likely because it is difficult to infer 99 PCs correctly without very large sample sizes when population structure is subtle ($F_{st} \leq 0.01$) but becomes easier when population structure is stronger (as in the $F_{st} = 0.1$ scenario). This is related to the known phase transition whereby it becomes possible to accurately infer latent factors (i.e., subpopulation membership) that

structure random matrices (e.g. SNP genotype matrices) once the sample size passes a certain threshold, depending on the strength of those latent factors⁷. In real-world genetic data, population structure likely exists on multiple scales (reflecting both recent and ancient structure) with genetic principal components only imperfectly capturing subtle and recent structure^{2,10}.

Since we simulated 100 subpopulations, we have simulated a less extreme scenario since the expected magnitude of bias due to population stratification decreases with the number of subpopulations. This is because the bias is due to the (finite sample) correlation between randomly sampled subpopulation allele frequencies and randomly sampled subpopulation phenotypic means. The expected correlation is zero, with the variance around zero (giving the magnitude of bias as measured by non-sampling variance) decreasing with the number of subpopulations. See the new material in the Methods section for more details. We appreciate that 50% of the phenotypic variance being explained by subpopulation membership is an extreme scenario. However, since the primary goal of these simulations is to examine potential bias-variance tradeoffs of different FGWAS estimators, we thought it necessary to simulate an extreme level of population stratification since the expected biases in FGWAS estimators are very small. As these simulations with 100 subpopulation show, the bias becomes so small that we are unable to detect it using 20,000 SNPs despite the large fraction of phenotypic variance explained by subpopulation mean differences. Simulating a lower level of population stratification would likely simply rescale the y-axis of our plots and make it hard to statistically detect any biases in the FGWAS estimators.

We have also shown that PC analysis is unlikely to be sufficient to fully adjust for population stratification bias in standard GWAS except in certain special scenarios, such as strong structure that can be accurately inferred. This is in line with existing literature demonstrating the inadequacy of genetic principal components to fully adjust for population stratification confounding when stratification is subtle or recent (see Zaidi and Mathieson for a nice discussion). By simulating a more realistic distribution of allele frequencies, we have shown that the family-based GWAS estimators do not exhibit increased bias when (ancestral) allele frequencies deviate from 0.5.

Why was robust estimator referred to with quotation marks?

We used quotation marks when first introducing an estimator to make it clear that we are naming the estimator as such. We are open to changing this if you or the editor thinks this is confusing, however.

COMPARISON OF ESTIMATORS

Can the authors justify why they quantified biases in terms of non-sampling variance? I thought that the statistical literature had more straightforward ways to quantify biases.

We thank the reviewer for pointing out that we did not make clear the reasoning behind evaluating bias in terms of non-sampling variance. We have now added a paragraph where we introduce the simulation results explaining our reasoning:

The bias from population stratification confounding is due to the (finite sample) correlation between the subpopulation allele frequencies and the subpopulation phenotypic means. Since the subpopulation allele frequencies and phenotypic means were sampled independently, the bias due to population stratification for an individual SNP has expectation zero (across repeated simulations) but has non-zero variance across SNPs (and repeated simulations) due to the finite number of subpopulations. The magnitude of population stratification confounding in the estimators can therefore be evaluated by their non-sampling variance — the variance in the estimates not explained by sampling error, which must be due to population stratification bias since there are no causal effects (Methods). We measure this relative to the non-sampling variance for standard GWAS and $F_{st} = 0.001$, which is comparable to the level of stratification bias that might be observed in a standard GWAS analysis of a homogeneous ancestry group. (We also give the mean Z^2 statistic, which should be 1 under the null. Mean values of Z^2 are common measures of test-statistic inflation in GWAS. However, the non-sampling variance provides a fairer comparison of levels of bias since the mean Z^2 is also affected by the size of the sampling errors, which varies across estimators.)

FIGURE 5: I was wondering if the behaviour reported in that figure was dependent on data dimension (i.e., number of individuals and SNPs). In particular, I'm curious to know if the bias observed for the unified estimator reflects some phase transition in the distribution of eigenvalues of the genotype matrix. This sort of behaviour is not uncommon and reflects inherent properties of random matrices. I'm not sure what would be the right experiment to conduct here but can the authors comment on that? Do they think that other situations could lead to large biases if even F_{st} is smaller?

For the Young et al. estimator, we have derived the bias due to population stratification (Supplementary Note). This depends upon the F_{st} and the magnitude of the bias, which is captured by the average non-transmitted coefficient from the regression of phenotype onto proband and parental genotype:

$$Y_i = \delta g_i + \alpha g_{\text{par}(i)} + \epsilon_i,$$

i.e. the value of α . For small values of F_{st} the bias in the Young et al. estimator is approximately $(F_{st}/2)\alpha$, so when F_{st} is small, the bias is negligible compared to the bias in standard GWAS (which is at least as large as α). The value of α is determined by the correlation of the parental genotype with the confounding factors. When F_{st} is small, α is also expected to be small since very little of the variation in genotype is between subpopulations, which is needed to capture population stratification confounding. However, α can also reflect confounding induced by assortative mating and IGEs, which do not require non-zero F_{st} . In any case, the magnitude of any such bias is likely to be very small compared to standard GWAS, but could be substantial when both F_{st} and α are substantial, as we show in our simulation results. We recommend that when F_{st} is substantial, the robust estimator should be preferred, even though the Young et al. and unified estimators may produce negligible bias when α is small (e.g. Figure 5). The unified estimator behaves similarly to the Young et al. estimator, except it will be somewhat more sensitive to population stratification confounding in proportion to the size of the ‘unrelated’ sample that is included. We simulated scenarios where the unrelated sample is ~10x larger than

the related sample, but it is likely the population stratification confounding is somewhat stronger when this ratio is increased.

We agree with the reviewer that the phase-transition behaviour of random matrices is relevant to these discussions. However, we believe it is primarily of relevance to the efficacy of inferred genetic principal components in adjusting for population stratification, as shown in our new simulations described above. For the simulation with $F_{st} = 0.1$ and 100 subpopulations, it appears that the phase-transition has occurred enabling accurate inference of the 99 PCs needed to separate the 100 subpopulations and perfectly correct for population structure. However, here the unified estimator displays no detectable bias without needing to adjust for PCs, suggesting this phase transition is not directly relevant for understanding the bias in the unified estimator.

APPLICATION TO REAL DATA

Can the authors directly quantify sample overlap (or relatedness) between UKB and MCS? What is the ancestry composition of MCS?

Directly quantifying the sample overlap or relatedness between UKB and MCS is unfortunately not possible due to data use agreements. It is possible that some parents in MCS are also in UKB, but it is likely only a handful if any at all. It is not possible for the MCS offspring to be in the UKB since the UKB sampled middle aged and older individuals, and the MCS sample was born around the year 2000.

The ancestry composition of MCS is described in this document (<https://cls.ucl.ac.uk/wp-content/uploads/2020/08/CLS-working-paper-2020-7-Collection-of-DNA-samples-and-genetic-data-at-scale-in-the-UK-Millennium-Cohort-Study.pdf>). Here's the main PCA figure displaying the genetic ancestry distribution of MCS projected onto 1kG principal components:

Figure 2. K-means clustering variance explained by clusters

Legend: MCS young people principle components one and two. Shapes define individual self-described ethnicity; the five colours define k-means continental ancestry. The eight self-described ethnicities are further plotted individually to aid visualization.

Note that we did not rely on self-reported ethnicity. We assigned samples to EUR or SAS clusters based on the procedure outlined in the methods:

“We projected MCS samples onto the top 20 principal components derived from 1000 Genomes data²³ using the OADP algorithm³⁷, and we determined the European and south Asian ancestry subsamples as those with 20 nearest neighbors in 1000 Genomes all part of the EUR and SAS superpopulations, respectively (see Supplementary Figure 5 for a visualization of the sample principal component distribution).”

As noted in the legend of Figure 7, this gave “European ancestry sample sizes: 5,285 for BMI, 5,285 for height, and 4,145 for EA. South Asian ancestry sample sizes: 685 for BMI and height, and 615 for EA.” As the plot above indicates, the south Asian ancestry samples are likely predominantly those identifying as Bangladeshi, Indian, and Pakistani ethnicity.

If I understand well Figure 7 then the PGI based on population effects has a larger accuracy in SAS than in EUR (at least for height). This seems to contradict what was previously reported in the literature. What could be the explanation for? Would the authors get the same effects if adjusting analyses in the MCS using principal components?

Thank you for highlighting this. We have performed a new analysis of MCS in response to this comment that adjusts for principal components. Now, we do not find that the population effect based PGIs have better prediction accuracy in SAS than in EUR, but we do still find that the direct-effect PGIs predict better in SAS than in EUR for height and BMI. We have rewritten the corresponding paragraph in the results:

The population-effect PGIs gave the most accurate predictions in the SAS sample for all phenotypes. However, for height, the best performing DGE PGI — from the unified estimator — performed nearly as well as the population-effect PGI, unlike in the EUR sample. As expected from previous analyses of cross-ancestry prediction^{17,31}, the prediction accuracy of the population-effect PGIs was lower in the SAS sample than in the EUR sample. In contrast, for height and BMI, the prediction accuracy for the DGE PGIs was higher in the SAS sample than in the EUR sample (Figure 7c). This difference was statistically significant for the unified estimator prediction on height ($\beta_{\text{SAS}} - \beta_{\text{EUR}} = 0.085$; S.E.=0.0392; P=0.0305, two-sided Z-test) (Supplementary Table 4). These results are consistent with DGE PGIs transferring better across ancestries, although the relatively small SAS sample precludes drawing strong conclusions.

Another quantity of interest for Figure 7 would be the regression slopes for each PGI. This is important to measure shrinkage of effect sizes and would give an indication about biases in real data.

We are not sure exactly what quantity the reviewer wants us to add here. We have plotted the correlations/partial correlations, which correspond to regression slopes with both PGIs and phenotypes standardized to have variance 1. Is there a different type of regression slope the reviewer would like to see displayed?

DISCUSSION

How would the robust estimator perform in an admixed population?

Thank you for this comment. It led us to formulating a new estimator that we have called the robust estimator and is robust to admixture and renaming the old robust estimator as the ‘non-transmitted estimator’ as detailed in the preamble for all reviewers.

Other comments

I could not see Table 3 in the main text. So I think there is a typo. Table 3 should be Table 2, right?

Yes, you are correct. Thanks for catching this.

The reader needs a bit more information about the phenotypes (at least in the UK Biobank). What field corresponds to the phenotypes used here. For example, I don’t think there is a UK Biobank field named “cognitive ability”. Can the authors specify which ones were used and whether only data from baseline were used if combined across visits.

We used the same phenotypes (with the same processing) as in Okbay et al. 2022, which we cited in the methods. However, we can see the utility of providing some more information on the phenotypes as suggested by the referee. We have added the following sentences: “Phenotypes were derived from baseline measurements. Note that ‘cognitive ability’ is derived from Field 20016 (‘Fluid intelligence score’).”

Can the authors report/mention the computational time of their methods somewhere in a table?

We have added a new supplementary table (Table S2) that gives some runtimes of our methods. We include a copy here for your convenience.

Table S2: Computational runtime of the four methods implemented in `snipar`.

Method	Sample size	Density	REML runtime (s)	Effect estimation runtime (s)
Sib-difference	41,209	5.233×10^{-5}	22.09	186.13
Robust	45,940	4.536×10^{-5}	23.76	544.85
Young 2022	39,525	5.294×10^{-5}	23.68	111.90
Unified	406,878	3.840×10^{-6}	173.93	844.91

Note. Runtimes for REML variance component estimation and SNP effect estimation (parallelized over 4 CPUs) are separately recorded for the four methods applied to UK Biobank height measurement and genotyping array data on chromosome 22 (10,911 SNPs). Sample size refers to the number of individuals with non-missing height measurements, whereas density refers to the number of nonzero entries relative to the total number of entries in the sparse GRM.

We report on this in the main text results section on the UK Biobank analyses:

“We give the example runtimes using 4 CPUs in Supplementary Table 2: 10,911 SNPs were analyzed with the sib-difference and Young et al. estimators in around 3 minutes or less, with the robust estimator in under 10 minutes, and with the unified estimator in around 14 minutes due to the much larger sample size.”

The table also shows that REML estimation of variance components takes on the order of 20 seconds for the related sample and 3 minutes for the unified estimator sample, and the results of this are shared across SNPs, so this is a trivial amount of the computation for a GWAS. These results show that it is feasible to analyze biobank scale samples with our software in a reasonable runtime, especially when taking advantage of multiple CPUs.

Reviewer #2:

Remarks to the Author:

Standard (population/non-family) GWAS provide the highest power for detecting genetic associations but effect estimates are impacted by assortative mating, population stratification and indirect genetic effects. Within-family GWAS provide better estimates of direct genetic effects but have much lower statistical power.

Guan et al provide an interesting technical report on improving the power of within-family GWAS by parental genotype imputation, expanding on previous work by the authors. In previously published work the authors presented an approach imputing parental genotypes for individuals with first degree relatives in the study.

In this work the approach has been extended to include two additional models: robust model - excluding IBD2 siblings where imputation was more challenging in previous work. unified model - including singletons (individuals without close relatives in the study).

The authors use simulations and theory to illustrate how improved statistical power across models comes with the risk of additional population stratification bias with the exception of the robust model. They then perform empirical analyses in UK Biobank (GWAS) and in the MCS (polygenic prediction).

The manuscript is generally well-written and although I haven't used the software package (and updates are coming), the software (via GitHub) looks polished and user-friendly. Please find my detailed comments below.

We thank the reviewer for their kind words on the manuscript, and we are glad that the reviewer found the manuscript to be well-written.

1)

I'm convinced that the robust model is preferable to the sibling differences model with higher statistical power (10-25%) at the cost of computational time for the imputation.

However, I'm not so convinced about the practical usefulness of the unified model because of potential susceptibility to population stratification. Generally researchers interested in within-family analyses are analysing phenotypes highly susceptible to non-direct genetic effects (height, educational attainment). Therefore, it seems to me that the unified estimator represents a compromise which has both advantages and disadvantages over the sib differences/robust and standard GWAS models. If going for statistical power, a researcher might prefer to use the standard GWAS and if going for less bias might prefer sib differences/robust estimators.

We are glad that the reviewer is convinced of the benefits of the robust estimator — note our reformulation of this estimator detailed in the preamble for all reviewers. We believe this to be an important innovation since it maximizes power for analysis of genetically diverse samples while being robust to confounding. As genomic data becomes more diverse, we anticipate as the robust estimator will become more widely applied. In fact, we are already working on applying this estimator to analyze the diverse data in the Mexico City Prospective Study.

However, when analyzing a relatively homogeneous subsample — such as used by the vast majority of previous GWASs — we believe the unified estimator is superior to the robust estimator and other alternatives. As shown in the initial submission, any population stratification bias in the unified estimator is undetectable for the scenarios with $F_{st} < 0.1$. The bias-variance tradeoff framework shows much greater precision for the unified estimator at effectively zero cost in terms of bias (in that it is undetectable) except for data with strong population structure. Furthermore, in our new simulations of more complex structure (100 subpopulations), the unified estimator did not have any detectable bias even when $F_{st} = 0.1$.

We do not see the unified estimator as a direct competitor with standard GWAS: standard GWAS estimates are output by *snipar* alongside the unified estimator estimates of direct genetic effects. We see them as complementary: one may prefer standard GWAS if the goal is to maximize out-of-sample prediction in similar contexts to the discovery sample, but one may prefer the unified estimator if the goal is some application that is sensitive to confounding (e.g. studying natural selection, estimating heritability and genetic correlation, Mendelian

randomization, etc.) or potentially prediction in different contexts such as within-family or across ancestries.

To help understand the strengths of the unified estimator and to provide a more measured summary:

a) Could the authors provide further empirical investigation of how the UKB results from the different models are affected by "non-direct" genetic effects? Maybe treating the sib difference/robust estimates as "the truth" and then comparing the GWAS effect sizes between the different models for height/educational attainment. Could look at top hits and random variants. In theory, the unified effect estimates should be on average the same but with lower standard errors. This would provide an empirical estimate of how close the unified estimator results are to the robust estimates in real data. If the "bias" is near-zero that would support the use of the unified estimator.

We thank the reviewer for highlighting what differences may exist between the direct genetic effect estimators in real data. The unified estimator and sib-difference/robust estimators use different samples, so we may not expect the estimands to be exactly the same. One important source of heterogeneity is that the robust and sib-difference estimators were applied to samples without ancestry restrictions, whereas the unified, standard GWAS, and Young et al. estimators were applied to the white British subsample. If effects differ between ancestries (e.g., due to gene-gene or gene-environment interactions), then we may expect to see systematic differences between effect estimates.

To investigate whether there was any systematic difference between effect estimates from the different estimators, we computed genetic correlations between the summary statistics of the different estimators for height and educational attainment (EA), now in Table S3. We include a copy here for your convenience:

Table S3: Genetic correlations among direct effect estimates from different estimators for height and EA in the UK Biobank data.

Methods	Height		EA	
	r_g	SE	r_g	SE
Young 2022 & unified	0.9990	0.0045	1.0329	0.0229
Young 2022 & sib-difference	1.0057	0.0064	1.0214	0.0366
Young 2022 & robust	1.0034	0.0053	0.9925	0.0265
Unified & sib-difference	0.9917	0.0084	1.0025	0.0534
Unified & robust	0.9816	0.0079	0.9331	0.0419
Sib-difference & robust	1.0145	0.0029	1.0672	0.0235

Note. Genetic correlations r_g (with standard errors SE) among direct effect estimates obtained from the Young 2022 method, the unified method, the sib-difference and the robust estimator are computed using LDSC with the default settings. We used the LD reference panel computed from phased haplotypes for the UK Biobank genotyping array SNPs.

Almost all genetic correlation estimates were very close to 1. We observed an unusually high genetic correlation estimate between sib-difference and robust estimators on EA, but this is probably due to the mismatch between LD-reference panel (European ancestry) and the mixed ancestries analyzed by the sib-difference and robust estimators.

In conclusion, we did not see any evidence for systematic differences in effect estimates from the different direct genetic effect estimators.

b) Abstract. The abstract mentions "increasing effective sample size for direct genetic effects by 44.5%-106.7% compared to using genetic differences between siblings" which I assume is from the unified model. The results mentions a more modest "effective sample size of between 10.8% (height) and 27.4%" presumably from the robust model. Given the unified model is potentially susceptible to population stratification please clarify that the increase comes with risk of bias or present the robust model increases. Technically the standard GWAS model would increase effective sample size for direct genetic effects substantially (at a cost)!

We agree that the gains of the unified estimator come at a potential cost of bias due to population stratification. However, any such bias is undetectable except when analyzing samples with much stronger population structure than are typically used in GWAS. This is not true for standard GWAS, where biases due to population stratification are present even for relatively homogeneous samples ($F_{st} = 0.001$). To make sure the abstract accurately reflects where our novel estimators are useful, we have modified the relevant section of the abstract to read:

We perform FGWAS on 19 phenotypes in the UK Biobank (UKB). In the white British subsample, the unified estimator (including samples without genotyped relatives) increased effective sample size for DGEs by 46.9%-106.5% compared to using genetic differences between siblings. In an analysis without ancestry restrictions, the structure-robust estimator increased the effective sample size for DGEs by 10.3%-21.0% compared to using siblings alone.

This now clearly shows that the unified estimator gains are restricted to an analysis of the white British subsample, and the robust estimator gains are from an analysis without ancestry restrictions.

2) Genome-wide versus autosomal.

Could the authors clarify if their framework is only applicable to the autosomes?

In particular, the X chromosome contains ~850 protein-coding genes so is potentially important, but is often omitted from "GWAS". Perhaps out of scope but if straightforward (and necessary) would it be possible to extend the model/software for the X chromosome?

If not please acknowledge as a limitation.

We thank the reviewer for highlighting this potential limitation. It is relatively straightforward to extend the methodology to the maternally inherited X-chromosome since this follows normal laws of Mendelian inheritance. However, it would not be possible to apply these methods (or any family-based GWAS type methods) to other sex chromosomes. We have added a clarification on this to introduction where we first introduce the family-based GWAS regression equation:

“(The FGWAS regression equation here only applies to the autosome. While it would be possible to apply a similar approach to maternally inherited X-chromosomes, FGWAS analyses of other sex-chromosomes or mitochondria is not possible.)”

Minor comments:

1) Introduction: some minor grammatical tweaks

"Adjustment for genetic principal components and linear mixed models (LMMs) reduce" (reduces)

"Family-based GWAS uses" (use)

Thanks. We have made this change.

2) The 0.05 threshold for IBD sparsity. Could the authors please elaborate on how sensitive the results are to this threshold. How would results change with a lower threshold?

The threshold only makes a difference in how far down the relatedness distribution we model the correlations in the residuals. How much impact this will have depends upon the distribution of relatedness in the sample, and whether relatedness can be reliably inferred below the 0.05 threshold. Lowering the threshold below 0.05 can have a fairly large impact on the computation time due to reducing the sparsity of the relationship matrix, but likely has little impact on the estimates and their estimated standard errors in most cases.

We have added the following text to the Methods section on linear mixed model inference, alluding to the fact that users can specify the threshold themselves if they wish to use a threshold different from 0.05:

“We chose this threshold as it enabled accurate modeling of residual correlations between close relatives^{35,37} — which is of importance for FGWAS in related samples — without requiring prohibitive memory and computation time. However, users are able to specify a different threshold to the software.”

3) Imputed versus sequence data.

UK Biobank and other studies are performing WGS on study participants. I suspect WGS data wouldn't have much of an advantage over imputed data for this framework but interested for the author's thoughts.

We do not believe there to be much advantage to WGS data over high quality imputed variants. However, low quality imputed variants are not suitable for FGWAS analyses, so we think that the WGS data in UKB will be useful as it would improve the % of genomic variants that can be used for FGWAS. We are preparing a manuscript that explores the utility of WGS data for FGWAS but this is beyond the scope of the present manuscript.

4) Please ensure the GitHub README is updated to provide guidance on the different models that can be run to ensure a user-friendly experience.

We thank the reviewer for this suggestion. We plan to include an extensive tutorial along the lines of the existing *snipar* tutorial (<https://snipar.readthedocs.io/en/latest/tutorial.html>) to guide people through using the different estimators. We will ensure this is ready, and the software made as user-friendly as possible, before the publication of a final version of the manuscript.

We thank the reviewers again for their input, which has resulted in a substantial strengthening of our manuscript.

Reviewer 2 comments:

The authors have well-addressed my previous comments. The new updates to the manuscript such as the admixture-robust estimator look great. I have only a few minor comments.

We thank the reviewer for their input and we are glad that they like the admixture-robust estimator.

1. In the introduction you discuss different contributors to genetic association estimates. Could also mention selection bias relating to non-randomness of the dataset. Participation bias in UKB is one element which people have investigated. There's also case-only analyses (e.g. disease progression GWAS) where estimates can be distorted after stratifying on a heritable factor. Higher disease polygenic scores associated with lower mortality in cases is a classic collider example.

We agree that participation bias is an important topic that can affect both population effects from standard GWAS and direct genetic effects from FGWAS. We have added a sentence to the introductory paragraph

“Genome-wide association studies (GWASs) have identified thousands of associations between genetic variants and human phenotypes and diseases¹. Standard GWAS study designs estimate the association between a phenotype and an allele by regression of individuals' phenotypes onto the number of copies of the allele that they carry, with some adjustment for covariates. Multiple phenomena contribute to the associations estimated by standard GWAS, which we call 'population effects', as they reflect the genotype-phenotype association in the population²⁻⁵: causal effects of alleles — both of the tested variants and those in linkage disequilibrium (LD) with the tested variant— carried by the individual on the individual, called direct genetic effects (DGEs); effects of alleles in relative(s) through the environment, called indirect genetic effects (IGEs) or genetic nurture⁶; and effects of other genetic and environmental factors that the tested variant is correlated with due to population stratification and assortative mating (AM)^{3,4,6-10}. **Biased sampling can also affect population effect estimates¹¹.**”

2. Introduction: 'overestimation of heritability and the traits' shared genetic architectures.' Could it also be underestimation under some theoretical model? Maybe mis-estimation or biased estimates is safer.

We agree with the reviewer that underestimation of heritability and traits shared genetic architecture is a possible outcome of confounding in GWAS association, especially for estimation of genetic correlations. We have changed the statement to “biased estimates of heritability and the traits' shared genetic architectures”.

3. Spurious inferences of disease causes by mendelian randomisation'. Suggest

rephrasing to ‘spurious inferences in Mendelian randomisation analyses’. I can’t think of examples where population based MR has been shown to be biased by AM/pop strat/ IGE for a disease outcome. I think most examples were with ‘social/behavioural’ outcome phenotypes. For example, the reference is BMI/Height on Educational Attainment. Population based GWAS of complex disease benchmark very well for successful drug targets and Mendelian disease genes.

We agree that this is a fair point and we’ve changed the statement to “biased inferences from Mendelian Randomization”.

4. Rare variants: I’m trying to wrap my head around how different within-family models work for rare variants ($MAF < 0.01$). Figure 2B has the theoretical gain in effective N across the MAF frequency between different models. This is the relative power, assume the absolute power/precision of within-family GWAS models plummet with allele frequency as we require heterozygosity within-families? e.g. a variant has MAF of 0.0001 in $N = 50K$. This would have MAC of 10 and only a subset of relateds would have differences. Rare variants more likely to co-occur within families lowering heterozygosity so power may be even lower than expected? No major action required, interested for thoughts on how the models perform in the rare/ultra-rare space.

Figure 2B only applies when the imputation does not used phased data: when using phased data, there is no dependence of relative effective sample size on MAF when using phased data. In fact, the imputation becomes easier when imputing rare variants as shown in Figure 2B: phase information becomes less useful when a variant is rare as the probability that the sibling pair is IBD for the rare allele when both are heterozygous approaches 1 as MAF approaches zero (assuming random mating). The sampling error of FGWAS estimators based on phased data depends upon the heterozygosity in the same way as standard GWAS estimates of population effects: both are proportional to $1/(f(1-f))$, where f is the allele frequency. Power for FGWAS for rare variants will likely be limited except in very large samples with genotyped first-degree relatives. In inbred populations, the calculations could be different as you have a reduction in heterozygosity. We agree that rare variant analysis is an important topic for both standard GWAS and FGWAS, but detailed discussion of this is beyond the scope of our paper, which focuses on common variant analysis.